# Mixture-of-Mamba:
# Enhancing Multi-Modal State-Space Models
# with Modality-Aware Sparsity

## Abstract

State Space Models (SSMs) such as Mamba have recently emerged as efficient alternatives to Transformers for sequential modeling. However, existing SSM architectures remain dense and modality-agnostic, limiting their efficiency in multi-modal pretraining. We introduce **Mixture-of-Mamba (MoM)**, a simple yet powerful approach that brings modality-aware sparsity directly into the core state-space projections of the Mamba block. MoM is the first architecture to integrate structured, modality-specific specialization inside the state-space dynamics themselves, enabling conditional computation within sequence modeling operations rather than only around them. We systematically evaluate MoM across three multi-modal pretraining frameworks—**Transfusion** (interleaved text–image with diffusion loss), **Chameleon** (text–image with discrete tokens), and an extended **three-modality setting with speech**. Across all settings, MoM achieves **equivalent or better loss with 35–65% fewer FLOPs**, scales to **1.5B parameters**, and delivers **up to 3–4× efficiency gains** on non-text modalities, while maintaining competitive text performance. Our ablations further show that joint decoupling of input, intermediate, and output projections yields super-additive improvements, highlighting the architectural insights of modality-aware sparsity in SSMs. Taken together, our results establish **Mixture-of-Mamba** as the first method to extend sparse, modality-specific design principles into the SSM family. This not only broadens the architectural toolkit for efficient multi-modal pretraining beyond Transformers, but also demonstrates that sparsity inside SSM dynamics is a promising direction for scalable foundation models.

## 1 Introduction

Large-scale multi-modal pretraining has been dominated by Transformer-based architectures, where sparsity and modality-aware parameterization have proven essential for efficiency and scalability. From early dual-stream models such as ViLBERT (Lu et al., 2019) to more recent Mixture-of-Experts (MoE) designs (Fedus et al., 2022; Liang et al., 2024), a central idea has been to allocate modality-specific parameters that activate only when needed, reducing compute while enhancing specialization. However, these advances remain tied to Transformers, which rely on global attention and offer natural insertion points for expert routing.

By contrast, *State Space Models (SSMs)* such as Mamba (Gu and Dao, 2023) have recently emerged as a compelling alternative, offering linear-time complexity and strong performance in language and vision tasks. Yet, existing SSM architectures are *dense and modality-agnostic*, providing no mechanism to leverage sparsity or modality-specific specialization. Prior attempts at combining MoE with SSMs—such as MoE-Mamba (Pióro et al., 2024) and BlackMamba (Anthony et al., 2024)—do so only peripherally, interleaving dense Mamba blocks with sparsified MLPs. These designs are fundamentally *orthogonal* to ours: they modify auxiliary components, while leaving the recurrent state-space dynamics untouched.

In this work, we present **Mixture-of-Mamba**, a new architecture that extends structured sparsity to the State Space Model (SSM) family. While *modality-specific* parameterization has been studied in Transformer-based architectures, applying these ideas within SSMs poses distinct challenges: Mamba is a recurrent,

linear-time sequence model without attention, leaving no obvious routing mechanism. Our key contribution is to introduce modality-aware sparsity directly inside the Mamba block by decoupling its core projection components according to modality. This design contrasts with prior MoE-SSM hybrids such as MoE-Mamba (Pióro et al., 2024) and BlackMamba (Anthony et al., 2024), which apply sparsity only to surrounding MLP layers while leaving the Mamba dynamics dense. Importantly, their approaches are orthogonal to ours: one can still apply peripheral MoE augmentation on top of Mixture-of-Mamba, but our method is the first to integrate structured sparsity within the state-space dynamics themselves. This enables specialization in the sequence modeling operations rather than only around them. Beyond architectural novelty, we provide the first systematic study of modality-aware sparsity in SSMs, evaluating across diverse multi-modal pretraining setups (e.g., Transfusion, Chameleon, three-modality) and demonstrating scalability up to 1.5B parameters.

We evaluate Mixture-of-Mamba across three challenging *multi-modal pretraining* settings: *Transfusion* (interleaved text and image with diffusion loss), *Chameleon* (interleaved text and discrete image tokens), and a new *three-modality extension* incorporating speech. Across all settings, MoM achieves equivalent or better validation loss with significantly fewer FLOPs, scaling efficiently up to 1.5B parameters. In the Transfusion setting, MoM matches dense Mamba's performance while requiring only 34.8% of the FLOPs at the 1.4B scale. In the three-modality setting, MoM reduces FLOPs by up to 75% while preserving competitive speech, text, and image performance. Our ablations further reveal that jointly decoupling multiple projections yields *super-additive improvements*, offering new insights into the design space of sparse SSMs.

**Contributions.** This work makes the following contributions:

- **Architectural novelty.** We present the first method to integrate modality-aware sparsity directly into the state-space dynamics of Mamba, complementing and orthogonal to prior MoE-SSM approaches that only sparsify peripheral MLP layers.
- **Empirical validation.** We provide the first comprehensive evaluation of modality-aware sparsity in SSMs, spanning Transfusion, Chameleon, and a three-modality pretraining setup. Our results demonstrate substantial efficiency gains and scalability to 1.5B parameters.
- **Ablation insights.** We show that jointly decoupling input, intermediate, and output projections yields super-additive improvements, highlighting new principles for designing sparse SSMs.

Taken together, our findings establish Mixture-of-Mamba as a simple, effective, and general design principle for efficient multi-modal pretraining. By extending sparsity from Transformers to SSMs, we open up new opportunities for scalable foundation models that combine the efficiency of state-space architectures with the specialization benefits of sparse computation.

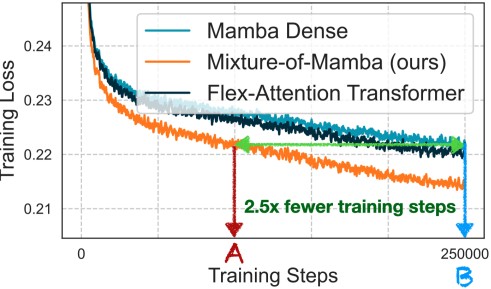

Figure 1: **Multi-modal pretraining on interleaved text and image data (Transfusion setting).** Validation loss on the **image modality** for 1.4B-parameter models. Mixture-of-Mamba not only converges faster than dense Mamba, but also outperforms the Flex-Attention Transformer baseline by a clear margin, reaching the same loss with 2.5× fewer training steps. This highlights the efficiency and competitiveness of integrating modality-aware sparsity directly into SSMs under a state-of-the-art multi-modal pretraining framework.

## 2 Mixture-of-Mamba for Efficient Multi-Modal LLM Pretraining

### 2.1 Modality-Aware Sparsity in Mamba

The key novelty of Mixture-of-Mamba lies in integrating *modality-aware sparsity* directly into the Mamba block. By dynamically selecting modality-specific parameters for each input token based on its modality, our approach enables Mamba to efficiently process interleaved multi-modal sequences (e.g., text and image tokens) while preserving computational efficiency.

For interleaved multi-modal tokens $\{x_1, x_2, \ldots, x_T\}$ from multiple modalities, such as text and image, modality-specific parameterization dynamically selects the appropriate parameters for each token during

processing. This general approach can apply to a wide range of transformations, such as linear, convolution, and activation-based transformations. In Mamba, which primarily relies on linear transformations, the approach takes the form:

$$f = Wx \quad \text{becomes} \quad f = \begin{cases} W_{\text{image}}x & \text{if } x \text{ is an image token} \\ W_{\text{text}}x & \text{if } x \text{ is a text token} \\ W_{\text{speech}}x & \text{if } x \text{ is a speech token} \end{cases} \tag{1}$$

Here, $W_{\text{image}}, W_{\text{text}}$, and $W_{\text{speech}}$ are the modality-specific parameter matrices dynamically selected based on the modality of each token. The selection is driven by a modal mask, which is applied at the first embedding layer. Thus, no manual intervention is required during training or inference. This design preserves model flexibility while enabling efficient specialization. We note that while Mamba focuses on linear projections, the general technique of modality-aware sparsity can extend to other types of parameterized layers as well.

### 2.1.1 The Mixture-of-Mamba Block

The Mixture-of-Mamba block (Algorithm 1) builds on Mamba by dynamically applying modality-specific parameterization to key projections during input processing. This technique allows the block to handle interleaved multi-modal tokens more efficiently by leveraging modality-aware sparsity.

Each Mixture-of-Mamba block consists of input projection $W_{\text{in\_proj}}$, intermediate projections $W_{\text{x\_proj}}$ and $W_{\text{dt\_proj}}$, and output projection $W_{\text{out\_proj}}$, all parameterized by the token's modality using the general parameterization function $\mathcal{M}(X, W, b; M)$. The general form of the parameterization is:

$$\mathcal{M}(X, W, b; M) = \bigcup_{m \in M} \{X_m W_m + b_m\} \tag{2}$$

where $X_m$ denotes the subset of tokens belonging to modality $m$, and $W_m$ and $b_m$ are the modality-specific parameters for that subset. This dynamic selection is applied at every stage of processing.

In Mixture-of-Mamba, projections explicitly processing input features belonging to a single modality—such as $W_{\text{in\_proj}}$, $W_{\text{x\_proj}}$, and $W_{\text{out\_proj}}$—are decoupled using modality-specific parameters. However, components like Conv1D and state transitions $A$ remain shared, as they operate across aggregated multi-modal features or RNN-like states, where the notion of modality is less well-defined. Specifically, $A$ governs the state-space dynamics across modalities and acts over shared latent representations that are both temporally and spatially entangled. It is also used inside the `selective_scan` CUDA kernel, which is optimized and non-trivial to modify without affecting runtime stability or efficiency. Moreover, Conv1D captures local temporal structure, where modality alignment is inherently ambiguous and spatial locality dominates. Sparsifying this layer based on modality could introduce discontinuities in local context modeling. Overall, our design ensures computational efficiency while retaining modality-specific specialization.

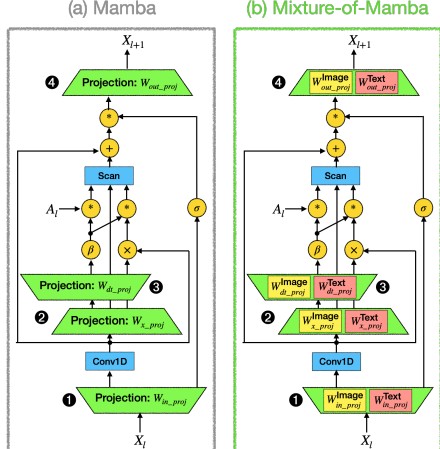

Figure 2: **Comparison of (a) the original Mamba block and (b) the proposed Mixture-of-Mamba block.** Mixture-of-Mamba applies *modality-specific parameterization* to all projection layers that directly process modality-dependent inputs: input projection (❶ $W_{\text{in\_proj}}$), intermediate projections (❷ $W_{\text{x\_proj}}$, ❸ $W_{\text{dt\_proj}}$), and output projection (❹ $W_{\text{out\_proj}}$). This design enables modality-aware sparsity within the state-space dynamics while preserving computational efficiency.

### 2.2 Multi-objective Training with Diffusion

Following Transfusion (Zhou et al., 2024), Mixture-of-Mamba is trained on interleaved multi-modal sequences of discrete text tokens and continuous image tokens using a combined objective that incorporates both language modeling and diffusion-based image generation. Each image is encoded as a sequence of latent patches using a Variational Autoencoder (VAE), where each patch is represented as a continuous vector. The

patches are sequenced left-to-right, top-to-bottom, and inserted into the discrete text sequence. The image latents are enclosed with the special tokens `<Begin of Image>` and `<End of Image>` to distinguish them from the text tokens.

The diffusion process follows the Denoising Diffusion Probabilistic Models (DDPM) (Ho et al., 2020), where Gaussian noise is progressively added to the latent image patches during the forward process. Given a clean latent patch $\mathbf{x}_0$, a noised version $\mathbf{x}_t$ at timestep $t$ is created as:

$$\mathbf{x}_t = \sqrt{\bar{\alpha}_t}\mathbf{x}_0 + \sqrt{1 - \bar{\alpha}_t}\boldsymbol{\epsilon}, \quad \boldsymbol{\epsilon} \sim \mathcal{N}(\mathbf{0}, \mathbf{I}), \tag{3}$$

where $\bar{\alpha}_t$ is determined by a cosine noise schedule (Nichol and Dhariwal, 2021), approximated as $\sqrt{\bar{\alpha}_t} \approx \cos(\frac{t}{T} \cdot \frac{\pi}{2})$ with adjustments. During training, noise is added to the latent patches at a randomly selected timestep $t$. The model is given the interleaved multi-modal sequence, where $\mathbf{x}_t$ replaces $\mathbf{x}_0$, and the objective is to predict the added noise $\boldsymbol{\epsilon}$. The overall training objective combines the autoregressive language modeling loss $\mathcal{L}_{\text{LM}}$, applied to the discrete text tokens, with the diffusion loss $\mathcal{L}_{\text{DDPM}}$, applied to the latent image patches, where $\lambda$ balances the contributions of the two losses:

$$\mathcal{L} = \mathcal{L}_{\text{LM}} + \lambda \cdot \mathcal{L}_{\text{DDPM}}. \tag{4}$$

Importantly, the conditioning for image generation is naturally embedded within the interleaved sequence. When denoising image patches during inference, the text prompt and the current

---

**Algorithm 1** Mixture-of-Mamba block

**input** $F_{in}$: Input sequence — $[b, l, f]$
$\quad A$: State transition matrix — $[d, n]$
$\quad W_{in\_proj}$: Input projection — $[f, 2d]$
$\quad W_{x\_proj}$: Intermediate projection — $[d, r + 2n]$
$\quad W_{dt\_proj}$: Intermediate projection — $[r, d]$
$\quad W_{out\_proj}$: Output projection — $[d, f]$
$\quad b$: Bias term — $[d]$
$\quad M$: Modality tag, one of $\{text, image, speech\}$
**output** $F_{out}$
1: $x, z \leftarrow \mathcal{M}(F_{in}, W_{in\_proj}; M)$
2: $u \leftarrow \text{SiLU}(\text{Conv1D}(x))$                   ▷ [b,ℓ,d]
3: $\delta, B, C \leftarrow \mathcal{M}(u, W_{x\_proj}; M)$      ▷ [b,ℓ,(r,n,n)]
4: $\Delta \leftarrow \log(1 + \exp((\mathcal{M}(\delta, W_{dt\_proj}, b; M))))$
5: $\overline{A} \leftarrow \exp(\Delta * A)$                  ▷ [b,ℓ,d,n]
6: $\overline{B} \leftarrow \Delta * (u \times B)$            ▷ [b,ℓ,d,n]
7: $h = 0$                             ▷ [b,d,n]
8: **for** $i = 0...\ell - 1$ **do**
9: $\quad h = h * \overline{A}_i + \overline{B}_i$         ▷ [b,d,n]
10: $\quad y_i = h \cdot C_i$               ▷ [b,d]
11: **end for**
12: $o \leftarrow (y + u) * \text{SiLU}(z)$
13: $F_{out} \leftarrow \mathcal{M}(o, W_{out\_proj}; M)$
14:
15: **function** $\mathcal{M}(X, W, b = \text{None}; M)$
16: $\quad$ **for** each modality $m \in M$ **do**
17: $\quad\quad I_m \leftarrow \{i : m_i = m\}$
18: $\quad\quad X_m \leftarrow \{x_i : i \in I_m\}$
19: $\quad\quad Y_m \leftarrow X_m W_m + b_m$
20: $\quad$ **end for**
21: $\quad$ return $Y \leftarrow \cup_{m \in M} Y_m$
22: **end function**=0

---

image latent $x_t$ serve as context to predict the noise for that step. This unified approach enables Mixture-of-Mamba to leverage the modality-aware sparsity to efficiently model both local intra-image dependencies and long-range inter-modal relationships across the sequence.

## 2.3 Training with Uniform Representations

As an alternative to the multi-objective training paradigm, we explore a unified representation strategy in which both text and image modalities are represented as discrete tokens. Following the Chameleon framework (Chameleon Team, 2024), we treat the image data as sequences of discrete tokens obtained through a pre-trained VQ-VAE (Gafni et al., 2022). Specifically, each image is encoded into a fixed number of tokens (e.g., 1,024) by quantizing its latent features into a learned codebook. These tokens are then arranged sequentially, similar to text tokens, resulting in a uniform discrete representation across both modalities.

During training, both text and image tokens are processed using the same autoregressive objective, where the model learns to predict the next token in the sequence given all previous tokens. Formally:

$$\mathcal{L}_{\text{uniform}} = \mathbb{E}_{\mathbf{x}_{1:T}} \left[ -\log P(\mathbf{x}_t \mid \mathbf{x}_{1:t-1}) \right], \tag{5}$$

where $\mathbf{x}_{1:T}$ represents the interleaved sequence of text and image tokens. This objective allows the model to treat text and image data equivalently, unifying the training process across modalities while relying solely on an autoregressive loss. The use of discrete tokens for images simplifies the training procedure by removing the need for separate loss formulations, as in the diffusion-based approach. It also aligns with the inherent sequence-to-sequence nature of Mixture-of-Mamba, where the same modality-aware sparsity design can be applied seamlessly across the discrete text and image tokens.

**Motivation and Robustness Testing.** We include this alternative strategy to evaluate the robustness of our Mixture-of-Mamba architecture under different choices of training objectives and data representations. By experimenting with uniform discrete representations, we demonstrate that Mixture-of-Mamba consistently

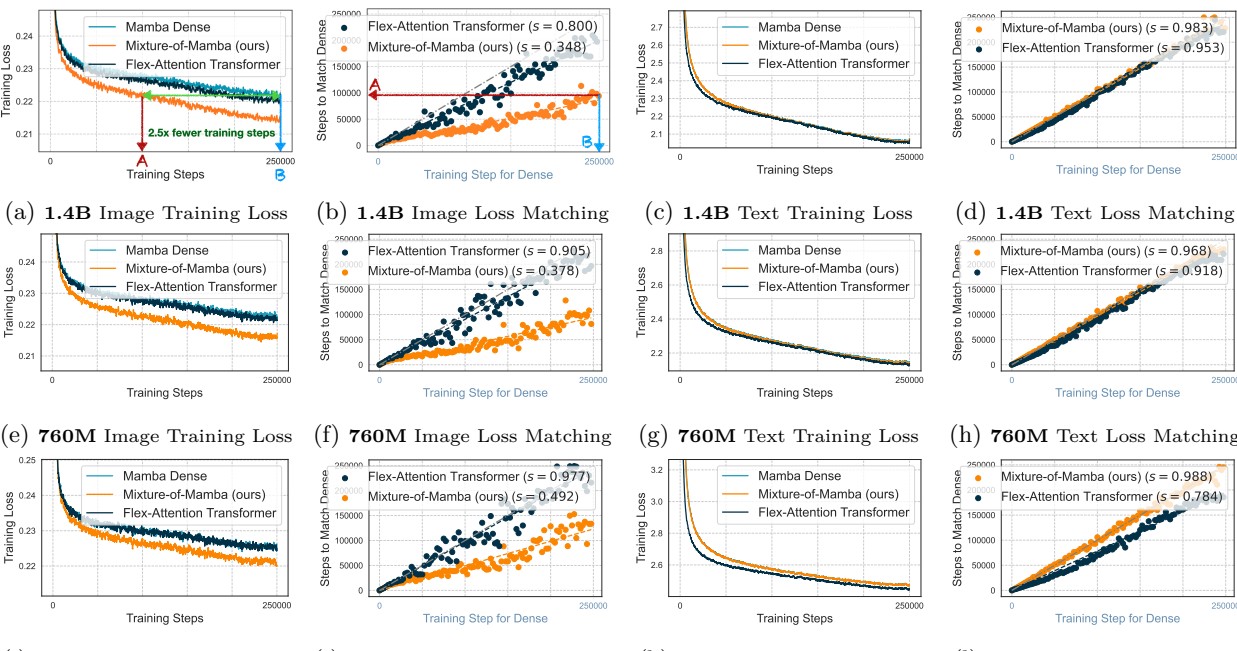

Figure 3: **Multi-modal pretraining in the Transfusion setting on interleaved text and image data across different model scales.** **(a, e, i)** Image training loss shows significant improvements for Mixture-of-Mamba (orange) over Mamba Dense (cyan) and Flex-Attention Transformer (dark gray) across all scales. **(b, f, j)** Image loss matching compares the training dynamics and shows that Mixture-of-Mamba and Flex-Attention Transformer reach the same loss at earlier training steps. **(c, g, k)** Text loss shows Mixture-of-Mamba is better than Mamba Dense and on par with the Flex-Attention Transformer. **(d, h, l)** Text loss matching shows that Mixture-of-Mamba are more efficient training than Mamba Dense, using fewer steps to achieve comparable loss.

outperforms Mamba Dense models across various settings, including both continuous (multi-objective) and discrete (uniform) representations. This highlights the versatility of Mixture-of-Mamba and its ability to deliver performance gains regardless of the underlying choice of modality representations or training objectives.

## 3 Results

### 3.1 Results in Multi-objective Training (Transfusion)

We first evaluate Mixture-of-Mamba (MoM) against Mamba Dense and Flex-Attention Transformer[1] in the **Transfusion setting**, where pretraining is performed on interleaved text and image data across three model scales: **163M**, **760M**, and **1.4B**. See our training configuration in Appendix Table 7.

For clarity, we also report performance gain as the relative reduction in final validation loss compared to the dense Mamba baseline. We chose not to include large-scale transformer models (e.g., Flamingo (Alayrac et al., 2022), Chameleon (Chameleon Team, 2024)) in our pretraining results due to significant differences in model scale and training setup, which would make direct comparisons less informative for our targeted architectural study. Our goal is to isolate and demonstrate the value of modality-aware sparsity in SSMs, and the current setup with Mamba Dense and Flex-Attention Transformer being the baselines achieves this fairly and effectively.

The detailed results are summarized in Table 1 and Figure 3, with further visualizations provided in Appendix Figures 4, 5, and 6. Relative training FLOPs reflect the computational cost required for MoM to match the training dynamics (similar loss) of Mamba Dense.

---

[1]Flex-Attention Transformer (i.e., Transfusion (Zhou et al., 2024)) combines both attention patterns by applying causal attention to every element in the sequence and bidirectional attention to images. This makes Flex-Attention Transformer an overestimated baseline for transformers because both Mamba and Mixture-of-Mamba are strictly causal across all elements, while Flex-Attention Transformer benefits from bidirectional attention within images.

Table 1: **Training and validation metrics across model scales in the Transfusion setting.** Mixture-of-Mamba consistently achieves competitive or superior performance in image metrics and maintains strong text performance compared to the baselines. The table also reports relative training FLOPs required for Mixture-of-Mamba and Flex-Attention Transformer to match Mamba's training dynamics, highlighting improved training efficiency. Best loss values in each row are shown in gray.

| Model Scale | Metric Category | Metric Name | Mamba Loss (↓) | Flex-Attention Transformer Loss (↓) | Mixture-of-Mamba Loss (↓) | Performance Gain over Mamba (%) (↑) | Relative Training FLOPs to Match Mamba (%) (↓) |
|---|---|---|---|---|---|---|---|
| 163M | Image Metrics | Training Loss | 0.2262 | 0.2250 | *0.2199* | 2.80% | 49.21% |
| | | CC12M Val. Loss | 0.2295 | 0.2293 | *0.2255* | 1.74% | 50.61% |
| | Text Metrics | Avg Training Loss | 2.4702 | *2.4424* | 2.4690 | 0.05% | 98.80% |
| | | C4 Val. Loss | 2.6917 | *2.6862* | 2.6912 | 0.02% | 99.88% |
| | | Wikipedia Val. Loss | 2.1884 | *2.1715* | 2.1870 | 0.06% | 99.81% |
| | Overall | Train Avg Loss | 3.6014 | *3.5674* | 3.5685 | 0.91% | 86.11% |
| 760M | Image Metrics | Training Loss | 0.2225 | 0.2213 | *0.2172* | 2.37% | 37.76% |
| | | CC12M Val. Loss | 0.2272 | 0.2253 | *0.2201* | 3.13% | 35.27% |
| | Text Metrics | Avg Training Loss | 2.1394 | *2.1253* | 2.1353 | 0.19% | 96.82% |
| | | C4 Val. Loss | 2.3593 | *2.3559* | 2.3555 | 0.16% | 99.01% |
| | | Wikipedia Val. Loss | 1.8191 | *1.8143* | 1.8149 | 0.23% | 99.11% |
| | Overall | Train Avg Loss | 3.2519 | 3.2318 | *3.2214* | 0.94% | 82.94% |
| 1.4B | Image Metrics | Training Loss | 0.2186 | 0.2221 | *0.2138* | 2.20% | 34.76% |
| | | CC12M Val. Loss | 0.2264 | 0.2247 | *0.2190* | 3.29% | 36.15% |
| | Text Metrics | Avg Training Loss | 2.0761 | *2.0673* | 2.0737 | 0.12% | 98.27% |
| | | C4 Val. Loss | 2.2726 | 2.2728 | *2.2695* | 0.13% | 99.34% |
| | | Wikipedia Val. Loss | 1.7205 | 1.7218 | *1.7164* | 0.24% | 99.30% |
| | Overall | Train Avg Loss | 3.1693 | 3.1777 | *3.1429* | 0.84% | 83.10% |

**Image Modality.** MoM consistently demonstrates superior performance in image modality training loss across all model scales. At the **1.4B** scale, MoM achieves a training loss of **0.2138**, outperforming Mamba Dense by **2.20%** while requiring only **34.76%** of the training FLOPs. Similar trends are observed at smaller scales: at the **760M** scale, MoM achieves a training loss of **0.2172**, a **2.37%** improvement over Mamba Dense, while reducing training FLOPs to **37.76%**.

The validation loss curves on the CC12M dataset (Table 1, Appendix Figure 5) further illustrate these trends. MoM consistently achieves lower image validation loss compared to Mamba Dense and Flex-Attention Transformer, with the improvements becoming more pronounced as model size increases. Additionally, loss matching curves demonstrate that MoM reaches equivalent loss values at earlier training steps, highlighting its improved training efficiency.

**Text Modality.** In the text modality, MoM consistently outperforms Mamba Dense across both training and validation metrics. At the **1.4B** scale, MoM achieves lower validation losses on both the C4 (**2.2695**) and Wikipedia (**1.7164**) datasets compared to Mamba Dense, despite their similar training losses. This indicates better generalization to unseen text data. Importantly, MoM also performs comparably to or better than Flex-Attention Transformer, particularly on validation losses, as shown in Appendix Figure 4. Similar trends are observed at smaller scales (**760M** and **163M**), where MoM reduces validation losses while maintaining high training efficiency. Loss matching results in Appendix Figure 4 (b, f, j) confirm that Mixture-of-Mamba aligns closely with or surpasses Mamba Dense, reaching comparable loss values earlier during training. These improvements highlight MoM's strong performance in text tasks while maintaining its computational efficiency.

**Overall Performance and Efficiency.** Across both image and text modalities, MoM consistently outperforms Mamba Dense in terms of loss reduction while requiring significantly fewer training FLOPs to achieve similar learning dynamics. At the **1.4B** scale, MoM improves the overall training loss by **0.84%** while requiring only **83.10%** of the training FLOPs. At smaller scales, such as **760M** and **163M**, MoM reduces the overall training loss by up to **0.94%**, while requiring just **82.94%** and **86.11%** of the FLOPs, respectively (Table 1, Appendix Figure 6). These results, summarized in Table 1 and Figure 3, and further supported by Appendix Figures 4, 5, and 6, underscoring MoM's effectiveness, scalability, and efficiency in the Transfusion setting.

Table 2: **Training and validation metrics across model scales in the Chameleon setting,** where both image and text modalities are represented as discrete tokens. MoM achieves substantial performance gain over Mamba Dense, with the **image modality** showing the largest gains. The **text modality** also exhibits significant improvements, in contrast to the Transfusion setting where text gains were more modest. The current table shows results for three model scales: **443M**, **880M**, and **1.5B**, due to space constraints. See Appendix Table 9 for the full results across all five model scales: **37M**, **94M**, **443M**, **880M**, and **1.5B**, which further highlight that MoM consistently achieves strong performance with reduced relative training FLOPs.

| Model Scale | Metric Category | Metric Name | Mamba Loss (↓) | Mixture-of-Mamba Loss (↓) | Performance Gain (%) (↑) | Relative Training FLOPs to Match Mamba (%) (↓) |
|---|---|---|---|---|---|---|
| 443M | Image Metrics | Training Loss | 5.3558 | 5.1703 | 3.46% | 33.40% |
| | | Obelisc Val. Loss | 4.5258 | 4.3546 | 3.78% | 35.10% |
| | | SSTK Val. Loss | 5.9179 | 5.7471 | 2.89% | 35.30% |
| | Text Metrics | Training Loss | 2.4637 | 2.3864 | 3.14% | 62.00% |
| | | Obelisc Val. Loss | 3.0544 | 2.9820 | 2.37% | 66.70% |
| | | SSTK Val. Loss | 2.7569 | 2.6250 | 4.78% | 54.70% |
| | Overall | Avg Training Loss | 3.6584 | 3.5364 | 3.33% | 47.90% |
| 880M | Image Metrics | Training Loss | 5.2260 | 5.1201 | 2.03% | 48.40% |
| | | Obelisc Val. Loss | 4.4127 | 4.3105 | 2.32% | 49.30% |
| | | SSTK Val. Loss | 5.7987 | 5.6986 | 1.73% | 50.50% |
| | Text Metrics | Training Loss | 2.3073 | 2.2438 | 2.75% | 65.60% |
| | | Obelisc Val. Loss | 2.8886 | 2.8313 | 1.99% | 72.80% |
| | | SSTK Val. Loss | 2.5483 | 2.4548 | 3.67% | 67.90% |
| | Overall | Avg Training Loss | 3.5130 | 3.4320 | 2.31% | 58.30% |
| 1.5B | Image Metrics | Training Loss | 5.1892 | 5.0591 | 2.51% | 42.50% |
| | | Obelisc Val. Loss | 4.3692 | 4.2510 | 2.71% | 44.50% |
| | | SSTK Val. Loss | 5.7546 | 5.6335 | 2.10% | 44.60% |
| | Text Metrics | Training Loss | 2.2284 | 2.1614 | 3.01% | 65.40% |
| | | Obelisc Val. Loss | 2.8020 | 2.7393 | 2.24% | 71.60% |
| | | SSTK Val. Loss | 2.4614 | 2.3455 | 4.71% | 62.10% |
| | Overall | Avg Training Loss | 3.4602 | 3.3670 | 2.69% | 54.70% |

## 3.2 Results in Training with Uniform Representations (Chameleon)

Next, we evaluate MoM in the **Chameleon setting**, where both image and text modalities are represented as discrete tokens. See our training configuration in Appendix Table 8. Results are summarized in Table 2, with full results across all five scales (**37M**, **94M**, **443M**, **880M**, and **1.5B**) provided in Appendix Table 9. Training dynamics and validation loss trends are visualized in Appendix Figures 7, 8, and 9.

**Image Modality.** MoM consistently demonstrates better performance in image modality training loss across all model scales, achieving substantial efficiency gains over Mamba Dense. At the **443M** scale, MoM achieves a training loss of **5.1703**, a **3.46%** improvement over Mamba Dense, while requiring only **33.40%** of the training FLOPs. Similar trends are observed at other scales: at the largest **1.5B** scale, MoM achieves a training loss of **5.0591**, a **2.51%** improvement, with only **42.50%** of the training FLOPs. At the smallest **37M** scale, MoM reduces training loss to **5.9561**, outperforming Mamba Dense by **2.85%** while requiring just **25.90%** of the FLOPs (Appendix Table 9). These results highlight MoM's ability to achieve improved performance and convergence efficiency consistently in the image modality across all model scales.

**Text Modality.** MoM demonstrates consistent improvements in text modality training loss across all model scales. At the largest **1.5B** scale, MoM reduces training loss to **2.1614**, a **3.01%** improvement over Mamba Dense, while requiring only **65.40%** of the training FLOPs. Validation loss on Obelisc and a proprietary version of the Shutterstock datasets (SSTK) exhibits similar trends, with MoM achieving notable improvements in loss values while maintaining significant efficiency gains (Appendix Figures 8 and 9). These results further highlight MoM's ability to deliver strong text performance with improved convergence efficiency.

## 3.3 Results in Training with Three Modalities (Chameleon + Speech)

Finally, to evaluate the robustness and scalability of MoM, we extend the Chameleon framework to include a third modality: speech, alongside image and text, with all modalities represented as discrete tokens. Speech

Table 3: **Training and validation metrics across model scales with three modalities: image, text, and speech.** This setting extends the Chameleon framework by incorporating **speech** beyond image and text, with all modalities represented as discrete tokens. MoM achieves consistent improvements over Mamba Dense across all scales, particularly in the **speech modality**, where performance gains reach up to **9.18%**. These gains are achieved with substantial reductions in training FLOPs, ranging from **10.30%** to **56.20%** relative to Mamba Dense. The results demonstrate that MoM generalizes effectively to a multi-modal setting with three modalities while delivering significant computational efficiency.

| Model Scale | Metric Category | Metric Name | Mamba Loss (↓) | Mixture-of-Mamba Loss (↓) | Performance Gain (%) (↑) | Relative Training FLOPs to Match Mamba (%) (↓) |
|---|---|---|---|---|---|---|
| 37M | Speech Metrics | Training Loss | 1.8159 | 1.6909 | 6.88% | 10.30% |
| | | LL60K Val. Loss | 1.6756 | 1.5217 | 9.18% | 13.60% |
| | | PPL30K Val. Loss | 1.8147 | 1.6845 | 7.17% | 13.60% |
| | Overall Metrics | Avg Training Loss | 4.2299 | 4.0759 | 3.64% | 45.00% |
| 94M | Speech Metrics | Training Loss | 1.6911 | 1.5662 | 7.38% | 11.90% |
| | | LL60K Val. Loss | 1.5235 | 1.3747 | 9.76% | 14.80% |
| | | PPL30K Val. Loss | 1.6951 | 1.6152 | 4.71% | 12.60% |
| | Overall Metrics | Avg Training Loss | 3.7756 | 3.6371 | 3.67% | 43.10% |
| 443M | Speech Metrics | Training Loss | 1.5414 | 1.4313 | 7.14% | 19.20% |
| | | LL60K Val. Loss | 1.3466 | 1.2113 | 10.05% | 24.70% |
| | | PPL30K Val. Loss | 1.5634 | 1.4790 | 5.40% | 22.00% |
| | Overall Metrics | Avg Training Loss | 3.3317 | 3.2096 | 3.66% | 44.00% |
| 880M | Speech Metrics | Training Loss | 1.4902 | 1.4054 | 5.69% | 22.40% |
| | | LL60K Val. Loss | 1.2939 | 1.1757 | 9.13% | 30.10% |
| | | PPL30K Val. Loss | 1.5400 | 1.4619 | 5.07% | 24.30% |
| | Overall Metrics | Avg Training Loss | 3.2289 | 3.1571 | 2.22% | 54.30% |
| 1.5B | Speech Metrics | Training Loss | 1.4790 | 1.3940 | 5.75% | 24.80% |
| | | LL60K Val. Loss | 1.2592 | 1.1552 | 8.26% | 32.10% |
| | | PPL30K Val. Loss | 1.5200 | 1.4387 | 5.35% | 27.60% |
| | Overall Metrics | Avg Training Loss | 3.1507 | 3.0545 | 3.05% | 56.20% |

data is tokenized using an in-house tokenizer, a variant of DinoSR (Liu et al., 2024a), which extracts semantic tokens with a vocabulary size of 500, where each token corresponds to 40ms of audio content. Results are summarized in Table 3, with additional training dynamics and evaluation loss trends visualized in Appendix Figures 11, 12, 13, and 14.

**Speech Modality.** MoM achieves substantial improvements in speech modality training loss across all model scales. At the **443M** scale, MoM improves speech training loss by **7.14%** compared to Mamba Dense. To match the training loss achieved by Mamba Dense, MoM requires only **19.20%** of the training FLOPs, demonstrating significant efficiency gains. Similar trends hold at the largest **1.5B** scale, where MoM achieves a **5.75%** improvement in speech training loss and matches Mamba Dense's loss with just **24.80%** of the training FLOPs.

**Overall training loss** is consistently reduced across scales. At the **1.5B** scale, MoM lowers the overall training loss by **3.05%**. When targeting the same loss as Mamba Dense, MoM achieves this with a **56.20%** reduction in relative training FLOPs, highlighting its improved computational efficiency. Performance in the image and text modalities similarly shows consistent improvements in training and validation losses relative to Mamba Dense. Full results are presented in Appendix Figures 13 and 14, where MoM's robust performance across all three modalities is further validated.

## 3.4 Ablation Study on Decoupling Components

To better understand the design choices underpinning Mixture-of-Mamba, we conduct an ablation study on the Chameleon + Speech setting at the 443M scale. We evaluate the impact of decoupling four key components—$W_{\text{in-proj}}$ (❶), $W_{\text{x-proj}}$ (❷), $W_{\text{dt-proj}}$ (❸), and $W_{\text{out-proj}}$ (❹)—individually and in various combinations. This analysis enables us to test both individual and combined contributions to the model's overall performance.

The results show that decoupling components individually yields varying degrees of improvement, with performance gains ranging from **0.63%** ($W_{\text{out-proj}}$) to **1.22%** ($W_{\text{in-proj}}$). Interestingly, some components ($W_{\text{x-proj}}$ and $W_{\text{dt-proj}}$) exhibit minimal or even slightly negative impact when decoupled alone. However,

Table 4: **Ablation study on the Chameleon + Speech** to evaluate the impact of decoupling individual components (**1**, **2**, **3**, **4**) vs. their combinations. Decoupling all components (**1+2+3+4**, **MoM**) achieves the best performance with a **3.80%** gain over the Mamba baseline. Notably, the performance gain achieved by decoupling all components exceeds the sum of gains from decoupling each component individually, highlighting the synergistic effect of combined decoupling.

| Ablation Study | Avg Training Loss ($\downarrow$) | Performance Gain (%) ($\uparrow$) |
|---|---|---|
| **443M** Mamba (without ❶❷❸❹) | 3.3317 | 0% (baseline) |
| ❶ (decouple $W_{in\_proj}$) | 3.2916 | 1.22% |
| ❷ (decouple $W_{x\_proj}$) | 3.3580 | -0.79% |
| ❸ (decouple $W_{dt\_proj}$) | 3.3525 | -0.62% |
| ❹ (decouple $W_{out\_proj}$) | 3.3109 | 0.63% |
| ❶+❷ (decouple $W_{in\_proj}, W_{x\_proj}$) | 3.2780 | 1.64% |
| ❶+❸ (decouple $W_{in\_proj}, W_{dt\_proj}$) | 3.2687 | 1.93% |
| ❶+❹ (decouple $W_{in\_proj}, W_{out\_proj}$) | 3.2599 | 2.20% |
| ❷+❸ (decouple $W_{x\_proj}, W_{dt\_proj}$) | 3.3214 | 0.31% |
| ❷+❹ (decouple $W_{x\_proj}, W_{out\_proj}$) | 3.2829 | 1.49% |
| ❸+❹ (decouple $W_{dt\_proj}, W_{out\_proj}$) | 3.2509 | 2.48% |
| ❶+❷+❸ (not decoupling $W_{out\_proj}$) | 3.2593 | 2.22% |
| ❶+❷+❹ (not decoupling $W_{dt\_proj}$) | 3.2312 | 3.11% |
| ❶+❸+❹ (not decoupling $W_{x\_proj}$) | 3.2342 | 3.01% |
| ❷+❸+❹ (not decoupling $W_{in\_proj}$) | 3.2773 | 1.66% |
| ❶+❷+❸+❹ (Mixture-of-Mamba) | **3.2096** | **3.80%** |

Table 5: To strengthen our results, we evaluated Mixture-of-Mamba on MS-COCO image generation, following the Transfusion setup. Below, we report FID scores ($\downarrow$, lower is better), grouped by model size.

| Model | # Params | FID ($\downarrow$) |
|---|---|---|
| **> 1B** | | |
| DALL · E (Ramesh et al., 2021) | 12B | 27.50 |
| Transfusion | 7.3B | 16.80 |
| Chameleon | 7B | 29.60 |
| CogView (Ding et al., 2021) | 4B | 27.10 |
| **Mixture-of-Mamba** | **1B** | **22.68** |
| Dense Mamba | 1B | 26.81 |
| Flex-Attn Transformer | 1B | 27.75 |
| **500M − 1B** | | |
| **Mixture-of-Mamba** | **760M** | **23.92** |
| Dense Mamba | 760M | 28.76 |
| Flex-Attn Transformer | 760M | 28.19 |
| MaskMamba-XL (Chen et al., 2025) | 741M | 25.93 |
| **< 500M** | | |
| LAFITE (Zhou et al., 2022) | 226M | 26.94 |
| **Mixture-of-Mamba** | **163M** | **26.60** |
| Dense Mamba | 163M | 37.07 |
| Flex-Attn Transformer | 163M | 35.81 |

decoupling multiple components in combination leads to significantly larger gains. For example, decoupling $W_{\text{in-proj}}$ and $W_{\text{out-proj}}$ (❶+❹) achieves a **2.20%** improvement, while decoupling three components (❶+❷+❹) further increases the gain to **3.11%**.

Most importantly, decoupling all four components simultaneously (❶+❷+❸+❹, Mixture-of-Mamba) achieves the largest improvement, with a performance gain of **3.80%** over the Mamba baseline. This result highlights a key observation: the gain from decoupling all components together exceeds the sum of individual gains, demonstrating a synergistic effect. The combination of all decoupled projections enables better parameter allocation across modalities, leading to more efficient and effective learning. In summary, the ablation study confirms that the design of Mixture-of-Mamba is both effective and interdependent. Decoupling all key components simultaneously is important to achieving the observed substantial performance gains.

### 3.5 Downstream Performance

To strengthen our results, we evaluated Mixture-of-Mamba on MS-COCO image generation (Lin et al., 2014), following the Transfusion setup. Specifically, we generate 256×256 images in a zero-shot fashion on 30k randomly sampled prompts from the validation set. We use 250 diffusion steps with a CFG coefficient of 1. As shown in Table 5, Mixture-of-Mamba consistently achieves lower FID than other models of similar scale, outperforming Dense Mamba across all sizes. Notably, our 1B variant surpasses the 7B Chameleon model, highlighting the effectiveness of our architecture for efficient generation. To further demonstrate image quality, we include qualitative samples of Mixture-of-Mamba 1B generated images at Appendix 15.

## 4 Related Work

Large-scale multi-modal pretraining has been advanced by Transformer-based architectures such as ViLBERT (Lu et al., 2019), Flamingo (Alayrac et al., 2022), Kosmos (Peng et al., 2023), and Chameleon (Chameleon Team, 2024), where modality-specific parameterization and sparse Mixture-of-Experts (MoE) routing have become central design principles (Fedus et al., 2022; Liang et al., 2024; Shen et al., 2023). These approaches demonstrate the importance of sparsity and conditional computation for

efficiency and scalability, but remain confined to attention-based architectures. Recent work has extended SSMs to multi-modal tasks (Qiao et al., 2024; Zhao et al., 2024; Zhu et al., 2024; Liu et al., 2024b; Yan et al., 2024; Hu et al., 2024), showing their adaptability but preserving dense state-space dynamics. The most relevant MoE-SSM variants, MoE-Mamba (Pióro et al., 2024) and BlackMamba (Anthony et al., 2024), interleave dense Mamba blocks with sparsified MLP layers. In contrast, our work is the first to introduce *structured, modality-aware sparsity directly inside the Mamba block*, enabling specialization within the state-space dynamics themselves rather than only around them. This distinction makes Mixture-of-Mamba complementary and orthogonal to prior MoE-SSM designs, and extends sparse modeling principles to a fundamentally different family of sequence models. A more detailed discussion of related work is provided in Appendix B.

## 5  Conclusion

We presented **Mixture-of-Mamba**, the first architecture to integrate *modality-aware sparsity directly into the state-space dynamics* of Mamba. Unlike prior MoE-SSM approaches that sparsify only peripheral MLP layers, our design enables specialization within the core projections of SSMs, offering a simple yet general principle for efficient multi-modal pretraining. Empirically, Mixture-of-Mamba achieves substantial FLOP reductions while matching or surpassing dense Mamba across text, image, and speech settings, scaling effectively up to 1.5B parameters. Ablation studies further show that joint decoupling of multiple projections yields super-additive gains, providing new insights into sparse SSM design. Taken together, our results establish Mixture-of-Mamba as a complementary and orthogonal direction to existing MoE methods, broadening the architectural toolkit for scalable foundation models beyond Transformers.

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

# A  Impact Statement

This work introduces efficiency improvements in multi-modal machine learning systems through modality-aware sparsity techniques. The primary impact is computational efficiency - Mixture-of-Mamba reduces computational costs by up to 65% while maintaining or improving performance. This has positive environmental implications through reduced energy consumption and democratizes access to multi-modal AI systems by lowering computational resource requirements. While these advances could enable beneficial applications in education, accessibility, and human-computer interaction, we acknowledge they could also facilitate potentially concerning applications. We encourage the research community to consider appropriate guidelines for responsible deployment of such technologies.

# B  Related Work

## B.1  State-Space Models and Multi-Modal Extensions

State-space models (SSMs) (Gu et al., 2021; Gu and Dao, 2023; Lieber et al., 2024) have recently gained traction as computationally efficient alternatives to Transformers for sequential modeling. Mamba (Gu and Dao, 2023), in particular, demonstrates strong performance on single-modality tasks by leveraging linear time complexity and advanced gating mechanisms. Extending Mamba to multi-modal tasks remains an active research area.

In vision-language modeling, VLMamba (Qiao et al., 2024) and Cobra (Zhao et al., 2024) augment Mamba by incorporating LLaVA-style projection modules, enabling image features to be mapped into the token space of the Mamba model for sequence modeling. In the vision domain, Vision Mamba (Zhu et al., 2024) introduces bidirectional scanning by chaining forward and backward SSM blocks, while VMamba (Liu et al., 2024b) further enhances image patch processing with a 2D Selective Scan (SS2D) module that traverses patches across multiple scanning paths.

For diffusion-based models, works such as DiffuSSM (Yan et al., 2024) and Zigma (Hu et al., 2024) replace attention mechanisms with SSMs for image and video generation. Zigma introduces a zigzag scanning scheme to improve efficiency for sequential diffusion tasks, while other approaches (Mo and Tian, 2024; Fei et al., 2024) explore bi-directional SSM architectures. While these works highlight the flexibility of Mamba in generative tasks, they focus primarily on architectural modifications for specific domains rather than general multi-modal pretraining.

The most related work to ours is MoE-Mamba (Pióro et al., 2024) and Blackmamba (Anthony et al., 2024), which interleave Mamba blocks with MoE-augmented MLPs to introduce sparsity. However, these hybrid designs apply sparsity only to the MLP layers, leaving the dense Mamba block unmodified. In contrast, our proposed Mixture-of-Mamba integrates modality-aware sparsity directly into the Mamba block by decoupling its projection components, enabling specialized computations for different modalities. This general design complements existing methods and offers new opportunities for computationally efficient multi-modal pretraining.

## B.2  Sparse Architectures for Multi-Modal Pretraining

Model sparsity, particularly Mixture-of-Experts (MoE), has been extensively explored in Transformers to reduce computational cost (Jacobs et al., 1991; Eigen et al., 2013; Shazeer et al., 2017; Lepikhin et al., 2020; Fedus et al., 2022; Jiang et al., 2024). MoE selectively activates subsets of parameters for each input token, allowing the model to specialize in different aspects of the data. However, challenges such as expert imbalance, bi-level optimization, and load balancing remain prevalent (Shazeer et al., 2017; Lepikhin et al., 2020; Tu et al., 2022).

In multi-modal tasks, modality-aware sparsity has emerged as an effective strategy. Works such as VLMo (Shen et al., 2023), MoMA (Lin et al., 2024), and related approaches (Wang et al., 2022; Shen et al., 2022; Bao et al., 2022; Long et al., 2023; Shen et al., 2025) assign modality-specific experts to handle the unique statistical

properties of text, images, and other data types. This improves specialization while avoiding the complexities of learned routing mechanisms (Liang et al., 2022).

Transformer-based architectures have further extended sparsity into attention mechanisms (Wang et al., 2023; Shen et al., 2024a;b; Liu et al., 2024c; Shen et al., 2024c). CogVLM (Wang et al., 2023) applies sparse techniques on top of a pre-trained Vicuna-7B model but remains limited to generating text outputs. Concurrently, Playground v3 (PGv3) (Liu et al., 2024c) integrates DiT-style image transformers with a frozen LLaMA-3 backbone to achieve state-of-the-art performance in text-to-image generation.

Our work differs fundamentally in two key aspects. First, Mixture-of-Mamba introduces *modality-aware sparsity* into the Mamba block itself, generalizing sparse architectures beyond Transformers to SSMs. Unlike prior works that sparsify only the MLP or attention components, we decouple projection components of the Mamba block, enabling efficient and specialized computations across modalities. Second, Mixture-of-Mamba is trained from scratch for multi-modal generation tasks, unlike approaches like CogVLM and PGv3 that fine-tune pre-trained backbones.

Furthermore, our design is complementary to existing MoE techniques. Prior work (Liang et al., 2024) has demonstrated that MoE-based sparsification can be combined with sparse architectures like Mixture-of-Transformers to achieve additional gains. Similarly, Mixture-of-Mamba can serve as a versatile and computationally efficient solution, offering new pathways for scalable multi-modal pretraining.

## C  Training Details

We provide detailed information on datasets, preprocessing, optimization, and hardware to ensure reproducibility. Unless otherwise noted, hyperparameters largely follow prior work on Transfusion, Mamba, and Chameleon, and were not extensively tuned due to compute constraints.

### C.1  Datasets

- **Text.** We use C4, Wikipedia, and proprietary in-house corpora.

- **Images.** We use CC12M and a filtered subset of LAION-400M.

- **Speech.** We pretrain on large-scale open-source speech datasets, summarized in Table 6.

| Dataset | Modality | Hours | #Speech Tokens[†] | #Text Tokens |
|---|---|---|---|---|
| People's Speech (Galvez et al., 2021) | Speech-only | 16,404 | 1.2B | – |
| Voxpopuli (En) (Wang et al., 2021) | Speech-only | 23,166 | 1.6B | – |
| LibriLight (Kahn et al., 2020) | Speech-only | 55,308 | 4.0B | – |
| Multilingual LibriSpeech (En) (Pratap et al., 2020) | Speech+Text | 44,585 | 3.2B | 0.5B |
| Spotify (Clifton et al., 2020) | Speech+Text | 57,290 | 4.2B | 0.7B |

Table 6: Speech pretraining datasets. [†]Speech tokens obtained by converting audio to 500 semantic tokens at 25Hz (40ms/token).

### C.2  Preprocessing

Images are resized to $256 \times 256$ and encoded into $32 \times 32$ latent patches using a pretrained VAE. In the Chameleon setting, both text and images are tokenized with a VQ-VAE (vocab size 8192; 1024 tokens/image).

### C.3  Optimization and Hyperparameters

We use AdamW with $(\beta_1, \beta_2) = (0.9, 0.95)$ and weight decay 0.1. The learning rate follows a cosine schedule with 2000 warmup steps. Peak learning rates are scale-dependent: $1.0 \times 10^{-3}$ (163M), $5.0 \times 10^{-4}$ (760M), and $3.0 \times 10^{-4}$ (1.4B). We apply gradient clipping of 1.0 and train in `bfloat16` with automatic loss scaling.

### C.4 Hardware and Batch Sizes

Models are trained on NVIDIA H100 GPUs. Scale-specific batch sizes are provided in Appendix Tables 7–8. All experiments use data parallelism with gradient accumulation to fit large batch sizes.

| Model Size | Hidden Dim. | Layers | Heads | Seq. Length | Batch Size/GPU | GPUs | Tokens/Batch | Steps |
|---|---|---|---|---|---|---|---|---|
| 163M | 768 | 16 | 12 | 4,096 | 4 | 56 | 1,048,576 | 250,000 |
| 760M | 1,536 | 24 | 24 | 4,096 | 4 | 56 | 1,048,576 | 250,000 |
| 1.4B | 2,048 | 24 | 16 | 4,096 | 2 | 128 | 1,048,576 | 250,000 |

Table 7: **Architectural specifications and training configurations of models across different parameter scales (Transfusion setting).**

| Model Size | Hidden Dim. | Layers | Heads | Seq. Length | Batch Size/GPU | GPUs | Tokens/Batch | Steps |
|---|---|---|---|---|---|---|---|---|
| 37M | 256 | 4 | 8 | 4,096 | 2 | 64 | 524,288 | 160,000 |
| 94M | 512 | 8 | 8 | 4,096 | 2 | 64 | 524,288 | 160,000 |
| 443M | 1,024 | 24 | 16 | 4,096 | 2 | 64 | 524,288 | 160,000 |
| 880M | 1,536 | 24 | 24 | 4,096 | 2 | 64 | 524,288 | 120,000 |
| 1.5B | 2,048 | 24 | 16 | 4,096 | 1 | 128 | 524,288 | 120,000 |

Table 8: **Architectural specifications and training configurations of models across different parameter scales (Chameleon setting and Chameleon+Speech setting).**

| Model Scale | Metric Category | Metric Name | Mamba Loss (↓) | Mixture-of-Mamba Loss (↓) | Performance Gain (%) (↑) | Relative Training FLOPs to Match Mamba (%) (↓) |
|---|---|---|---|---|---|---|
| 37M | Image Metrics | Training Loss | 6.1308 | 5.9561 | 2.85% | 25.90% |
| | | Obelisc Val. Loss | 5.2866 | 5.1124 | 3.29% | 26.60% |
| | | SSTK Val. Loss | 6.6694 | 6.5023 | 2.51% | 27.50% |
| | Text Metrics | Training Loss | 3.6262 | 3.5175 | 3.00% | 60.90% |
| | | Obelisc Val. Loss | 4.1244 | 4.0469 | 1.88% | 64.80% |
| | | SSTK Val. Loss | 4.0417 | 3.9533 | 2.19% | 57.50% |
| | Overall | Avg Training Loss | 4.6607 | 4.5247 | 2.92% | 50.70% |
| 94M | Image Metrics | Training Loss | 5.7609 | 5.6057 | 2.69% | 35.70% |
| | | Obelisc Val. Loss | 4.9231 | 4.7683 | 3.14% | 35.30% |
| | | SSTK Val. Loss | 6.3130 | 6.1652 | 2.34% | 37.00% |
| | Text Metrics | Training Loss | 3.0294 | 2.9414 | 2.90% | 58.40% |
| | | Obelisc Val. Loss | 3.6016 | 3.5270 | 2.07% | 62.60% |
| | | SSTK Val. Loss | 3.4109 | 3.2901 | 3.54% | 61.40% |
| | Overall | Avg Training Loss | 4.1577 | 4.0419 | 2.78% | 49.80% |
| 443M | Image Metrics | Training Loss | 5.3558 | 5.1703 | 3.46% | 33.40% |
| | | Obelisc Val. Loss | 4.5258 | 4.3546 | 3.78% | 35.10% |
| | | SSTK Val. Loss | 5.9179 | 5.7471 | 2.89% | 35.30% |
| | Text Metrics | Training Loss | 2.4637 | 2.3864 | 3.14% | 62.00% |
| | | Obelisc Val. Loss | 3.0544 | 2.9820 | 2.37% | 66.70% |
| | | SSTK Val. Loss | 2.7569 | 2.6250 | 4.78% | 54.70% |
| | Overall | Avg Training Loss | 3.6584 | 3.5364 | 3.33% | 47.90% |
| 880M | Image Metrics | Training Loss | 5.2260 | 5.1201 | 2.03% | 48.40% |
| | | Obelisc Val. Loss | 4.4127 | 4.3105 | 2.32% | 49.30% |
| | | SSTK Val. Loss | 5.7987 | 5.6986 | 1.73% | 50.50% |
| | Text Metrics | Training Loss | 2.3073 | 2.2438 | 2.75% | 65.60% |
| | | Obelisc Val. Loss | 2.8886 | 2.8313 | 1.99% | 72.80% |
| | | SSTK Val. Loss | 2.5483 | 2.4548 | 3.67% | 67.90% |
| | Overall | Avg Training Loss | 3.5130 | 3.4320 | 2.31% | 58.30% |
| 1.5B | Image Metrics | Training Loss | 5.1892 | 5.0591 | 2.51% | 42.50% |
| | | Obelisc Val. Loss | 4.3692 | 4.2510 | 2.71% | 44.50% |
| | | SSTK Val. Loss | 5.7546 | 5.6335 | 2.10% | 44.60% |
| | Text Metrics | Training Loss | 2.2284 | 2.1614 | 3.01% | 65.40% |
| | | Obelisc Val. Loss | 2.8020 | 2.7393 | 2.24% | 71.60% |
| | | SSTK Val. Loss | 2.4614 | 2.3455 | 4.71% | 62.10% |
| | Overall | Avg Training Loss | 3.4602 | 3.3670 | 2.69% | 54.70% |

Table 9: **Training and validation metrics across model scales in the Chameleon setting.** In this setting, both image and text modalities are represented as discrete tokens. Mixture-of-Mamba achieves substantial performance improvements over Mamba Dense, with the **image modality** showing the largest gains across all five model scales: **37M**, **94M**, **443M**, **880M**, and **1.5B**. Notably, the **text modality** also exhibits significant improvements, in contrast to the Transfusion setting where text gains were more modest. These results further highlight the effectiveness and efficiency of Mixture-of-Mamba, which consistently achieves strong performance with reduced relative training FLOPs.

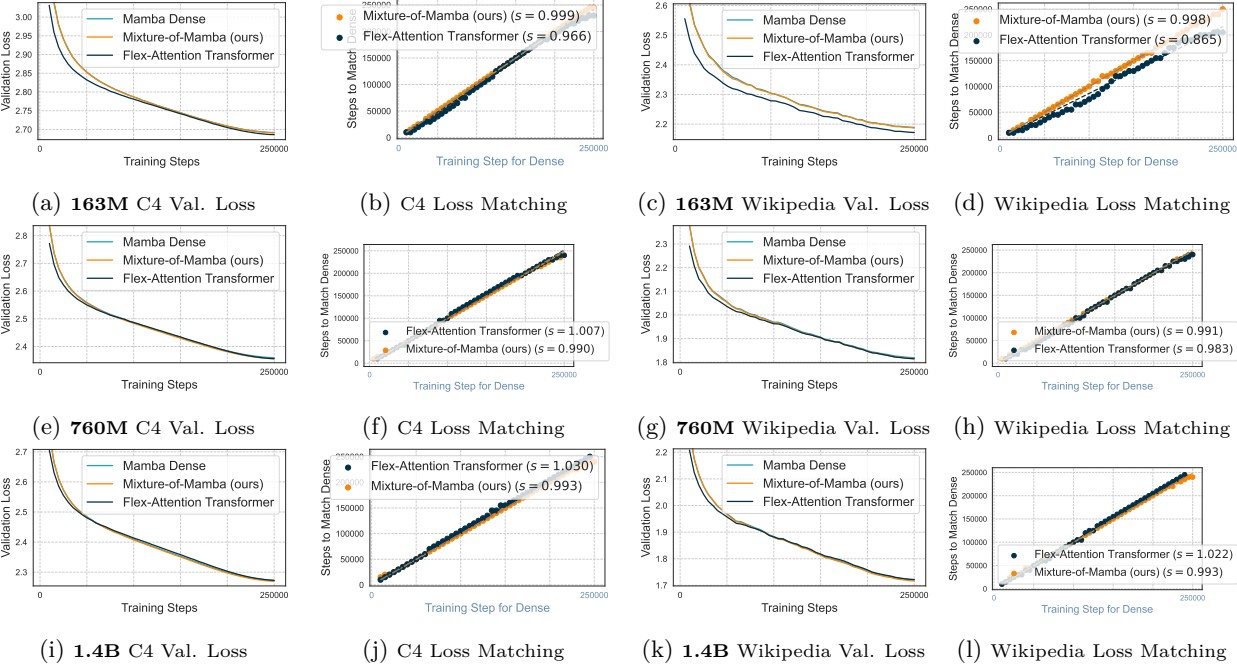

Figure 4: Validation loss and loss matching for text modality across model scales (**C4** and **Wikipedia** datasets) during multi-modal pretraining in the **Transfusion setting**. Results are shown for Mixture-of-Mamba, Mamba Dense, and Flex-Attention Transformer at three model scales: **163M**, **760M**, and **1.4B**. (**a, e, i**) Validation loss on the **C4 dataset** shows that Mixture-of-Mamba achieves comparable performance at **163M** and performs marginally better than Mamba Dense and Flex-Attention Transformer at the **760M** and **1.4B** scales. (**b, f, j**) Loss matching for C4 demonstrates that Mixture-of-Mamba reaches similar or slightly lower loss values at earlier training steps compared to Mamba Dense. (**c, g, k**) Validation loss on the **Wikipedia dataset** follows a similar trend, with Mixture-of-Mamba showing marginal improvements at the **760M** and **1.4B** scales. (**d, h, l**) Loss matching for Wikipedia illustrates efficient training dynamics, with Mixture-of-Mamba aligning closely with Flex-Attention Transformer while reaching comparable or slightly lower loss values than Mamba Dense. Overall, Mixture-of-Mamba demonstrates moderate improvements over both baselines at the larger scales (**760M** and **1.4B**).

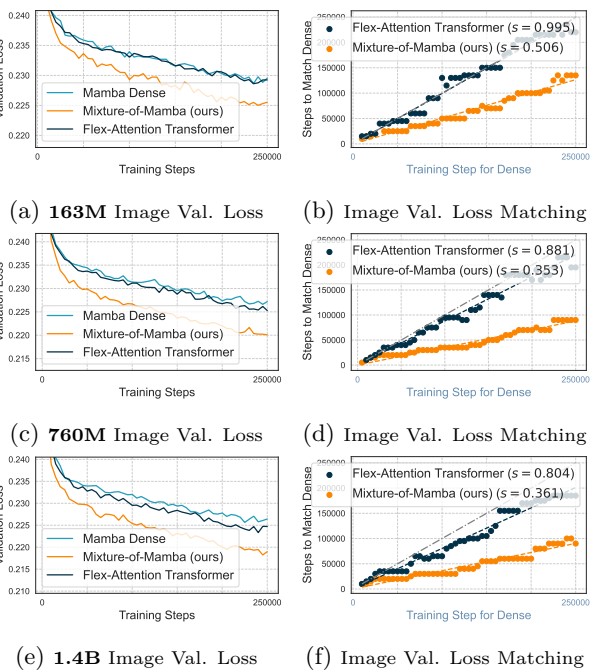

(a) **163M** Image Val. Loss    (b) Image Val. Loss Matching

(c) **760M** Image Val. Loss    (d) Image Val. Loss Matching

(e) **1.4B** Image Val. Loss    (f) Image Val. Loss Matching

Figure 5: Image validation loss and loss matching on the **CC12M dataset** across three model scales: **163M**, **760M**, and **1.4B** during multi-modal pretraining in the **Transfusion setting**. (**a, c, e**) Validation loss curves show that Mixture-of-Mamba achieves substantially lower image validation loss compared to Mamba Dense and Flex-Attention Transformer across all scales, with the improvement becoming more pronounced as model size increases. (**b, d, f**) Loss matching curves demonstrate that Mixture-of-Mamba reaches the same loss values at earlier training steps compared to Mamba Dense, highlighting improved training efficiency. Overall, Mixture-of-Mamba achieves large improvements in image validation loss on the **CC12M dataset**, showcasing its effectiveness in the image modality.

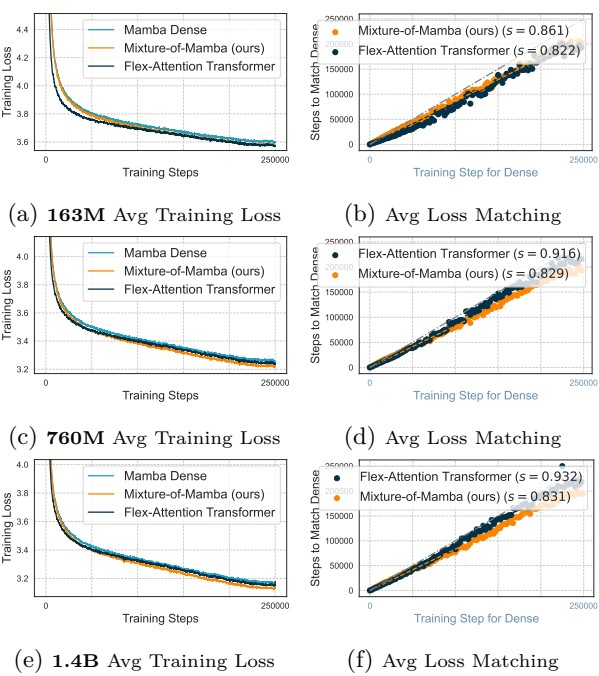

(a) **163M** Avg Training Loss  (b) Avg Loss Matching

(c) **760M** Avg Training Loss  (d) Avg Loss Matching

(e) **1.4B** Avg Training Loss  (f) Avg Loss Matching

Figure 6: Overall training loss and loss matching during multi-modal pretraining in the **Transfusion setting**. Results are shown for Mixture-of-Mamba, Mamba Dense, and Flex-Attention Transformer at three model scales: **163M**, **760M**, and **1.4B**. (**a, c, e**) Training loss averaged across the image and text modalities demonstrates that Mixture-of-Mamba achieves substantial improvements over Mamba Dense, with a notable reduction in training loss across all scales. (**b, d, f**) Loss matching results show that Mixture-of-Mamba and Flex-Attention Transformer reach the same loss values at earlier training steps compared to Mamba Dense, highlighting improved training efficiency. *Note:* The image loss in the Transfusion setting corresponds to the diffusion loss, which is of smaller magnitude compared to the cross-entropy loss in the text modality. Overall, Mixture-of-Mamba demonstrates significant gains in training loss and efficiency across multi-modal pretraining.

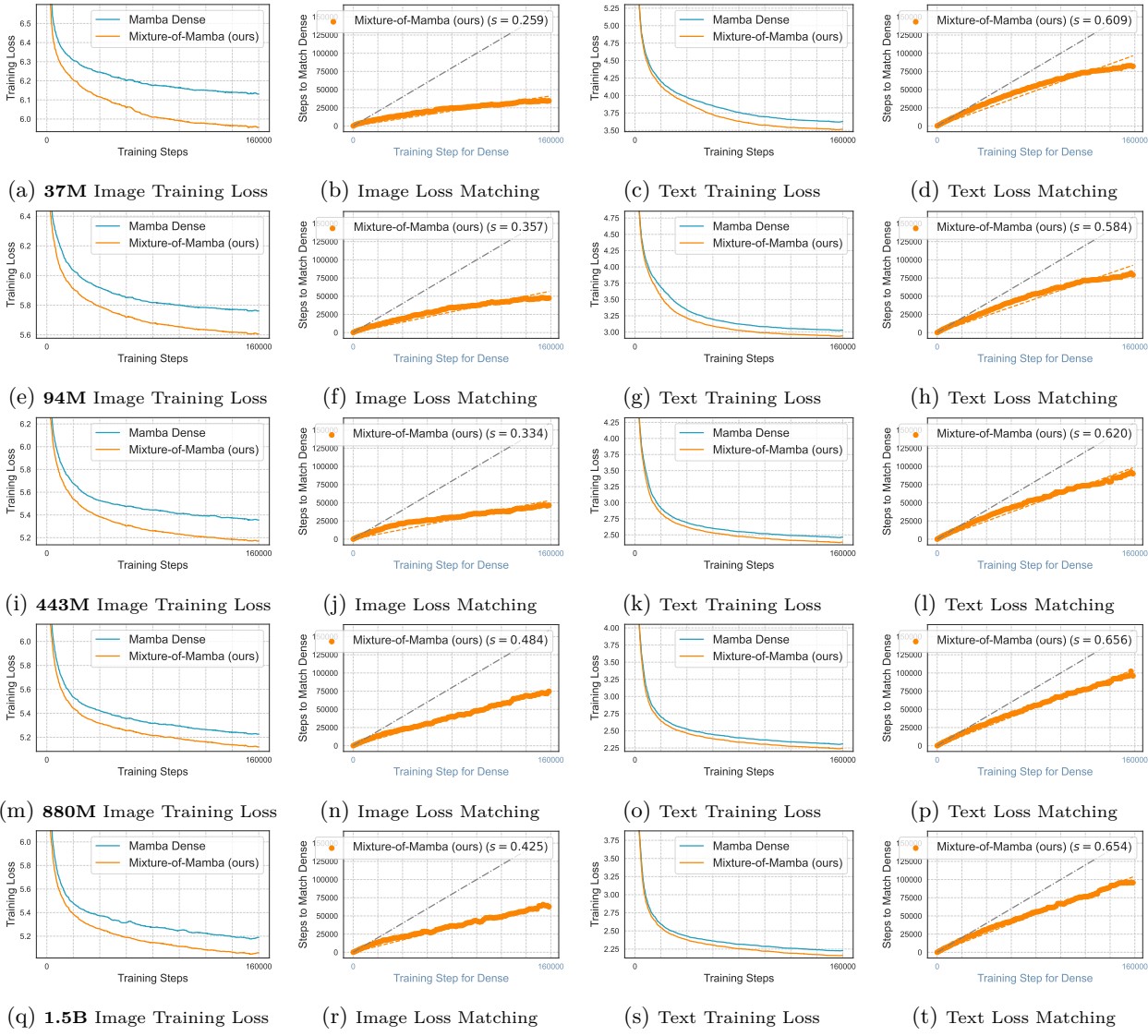

Figure 7: **Modality-specific pre-training loss and step matching plots across model scales (Chameleon setting).** Training loss and loss matching are reported for image and text modalities across five model scales: **37M**, **94M**, **443M**, **880M**, and **1.5B**. **(a, e, i, m, q)** Image training loss shows significant improvements for Mixture-of-Mamba (orange), which consistently achieves lower loss compared to Mamba Dense (cyan) across all scales. **(b, f, j, n, r)** Image loss matching compares the training dynamics and shows that Mixture-of-Mamba reaches the same loss values at earlier training steps compared to Mamba Dense, highlighting its improved efficiency. **(c, g, k, o, s)** Text training loss demonstrates competitive performance, with Mixture-of-Mamba achieving slightly lower loss values compared to Mamba Dense. **(d, h, l, p, t)** Text loss matching illustrates that Mixture-of-Mamba reaches the same loss values at earlier training steps compared to Mamba Dense, reflecting its efficient training dynamics. Overall, in the **Chameleon setting**, Mixture-of-Mamba achieves consistent improvements in the image modality, with substantial computational savings, while also demonstrating meaningful gains in the text modality.

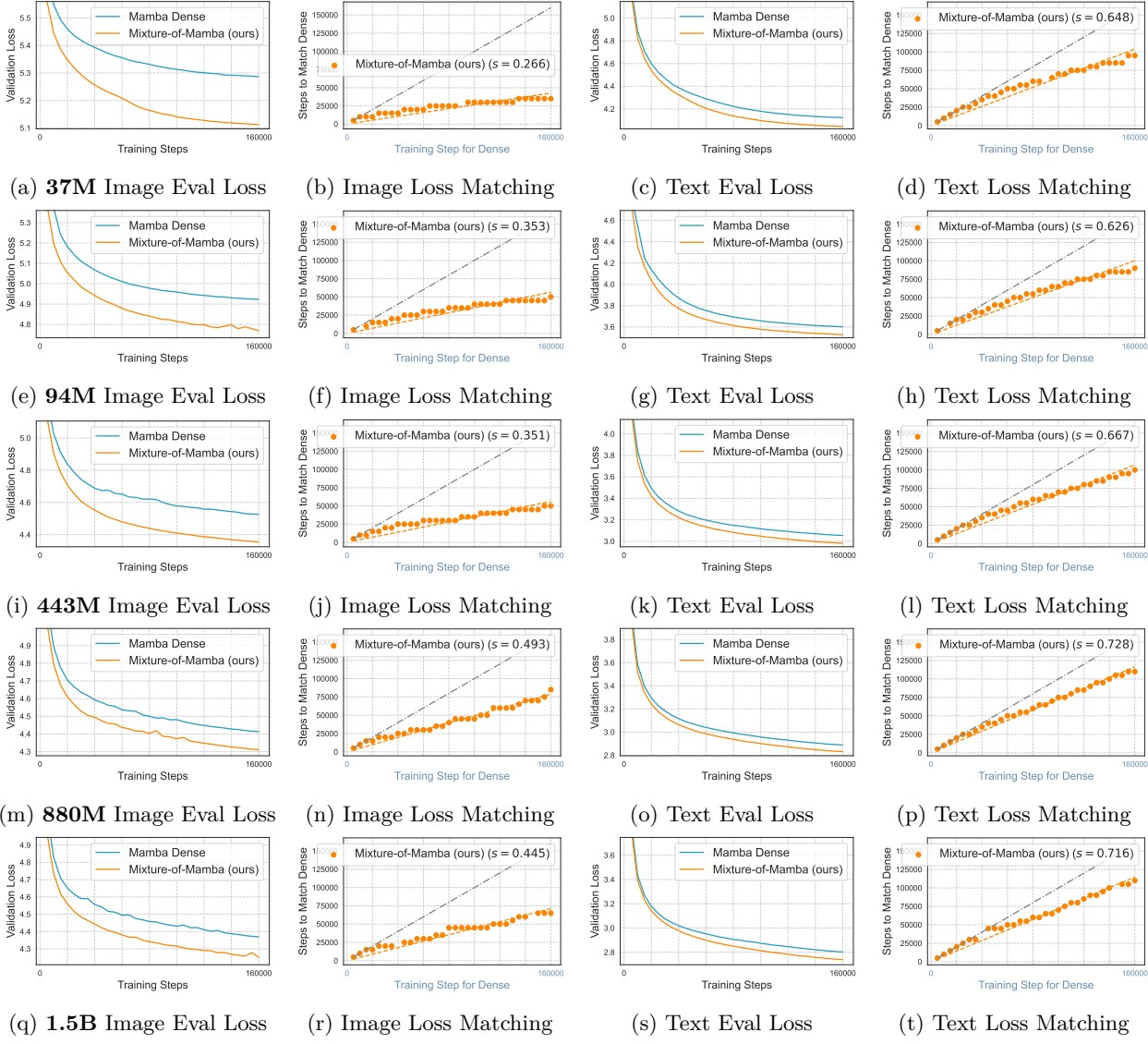

(a) **37M** Image Eval Loss    (b) Image Loss Matching    (c) Text Eval Loss    (d) Text Loss Matching

(e) **94M** Image Eval Loss    (f) Image Loss Matching    (g) Text Eval Loss    (h) Text Loss Matching

(i) **443M** Image Eval Loss    (j) Image Loss Matching    (k) Text Eval Loss    (l) Text Loss Matching

(m) **880M** Image Eval Loss    (n) Image Loss Matching    (o) Text Eval Loss    (p) Text Loss Matching

(q) **1.5B** Image Eval Loss    (r) Image Loss Matching    (s) Text Eval Loss    (t) Text Loss Matching

Figure 8: **Training and evaluation losses for image and text modalities across model scales in the Chameleon setting on the Obelisc dataset.** Results are shown for Mixture-of-Mamba and Mamba Dense across five model scales: **37M**, **94M**, **443M**, **880M**, and **1.5B**. **(a, e, i, m, q)** Image evaluation loss demonstrates consistent improvements for Mixture-of-Mamba (orange), achieving lower loss compared to Mamba Dense (cyan) across all scales. **(b, f, j, n, r)** Image loss matching shows that Mixture-of-Mamba reaches the same loss values at earlier training steps compared to Mamba Dense, reflecting its improved training efficiency. **(c, g, k, o, s)** Text evaluation loss indicates competitive results for Mixture-of-Mamba, achieving lower losses relative to Mamba Dense. **(d, h, l, p, t)** Text loss matching highlights that Mixture-of-Mamba reaches the same loss values at earlier training steps, further demonstrating its efficiency in the text modality. Overall, Mixture-of-Mamba achieves strong and consistent improvements in both image and text modalities across all model scales in the Chameleon setting evaluated on the Obelisc dataset.

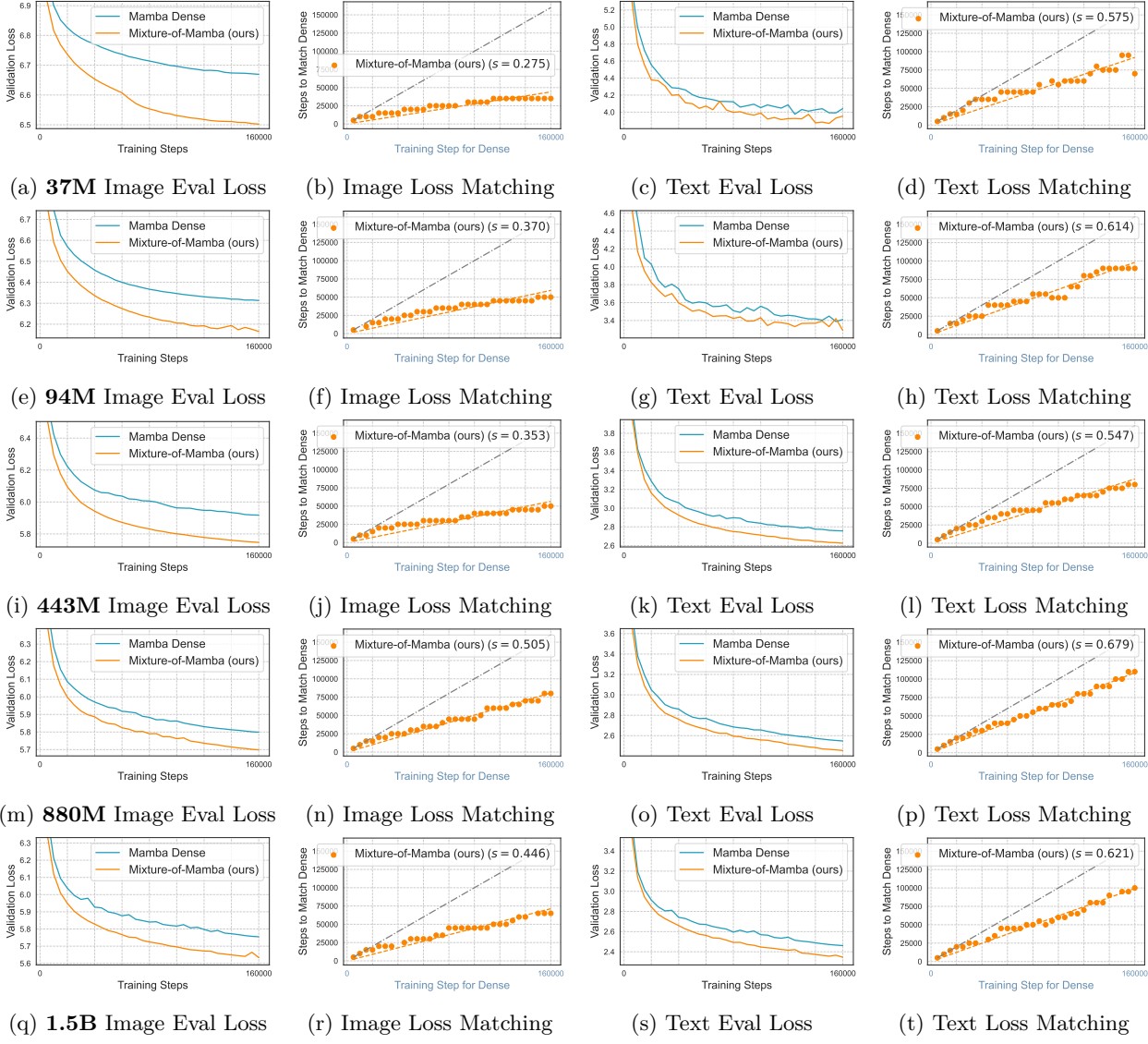

Figure 9: **Training and evaluation losses for image and text modalities across model scales in the Chameleon setting on the Shutterstock dataset.** Results are shown for Mixture-of-Mamba and Mamba Dense across five model scales: **37M**, **94M**, **443M**, **880M**, and **1.5B**. **(a, e, i, m, q)** Image evaluation loss demonstrates consistent improvements for Mixture-of-Mamba (orange), achieving lower loss compared to Mamba Dense (cyan) across all scales. **(b, f, j, n, r)** Image loss matching shows that Mixture-of-Mamba reaches the same loss values at earlier training steps compared to Mamba Dense, reflecting its improved training efficiency. **(c, g, k, o, s)** Text evaluation loss indicates competitive results for Mixture-of-Mamba, achieving lower losses relative to Mamba Dense. **(d, h, l, p, t)** Text loss matching highlights that Mixture-of-Mamba reaches the same loss values at earlier training steps, further demonstrating its efficiency in the text modality. Overall, Mixture-of-Mamba achieves strong and consistent improvements in both image and text modalities across all model scales in the Chameleon setting evaluated on the Shutterstock dataset.

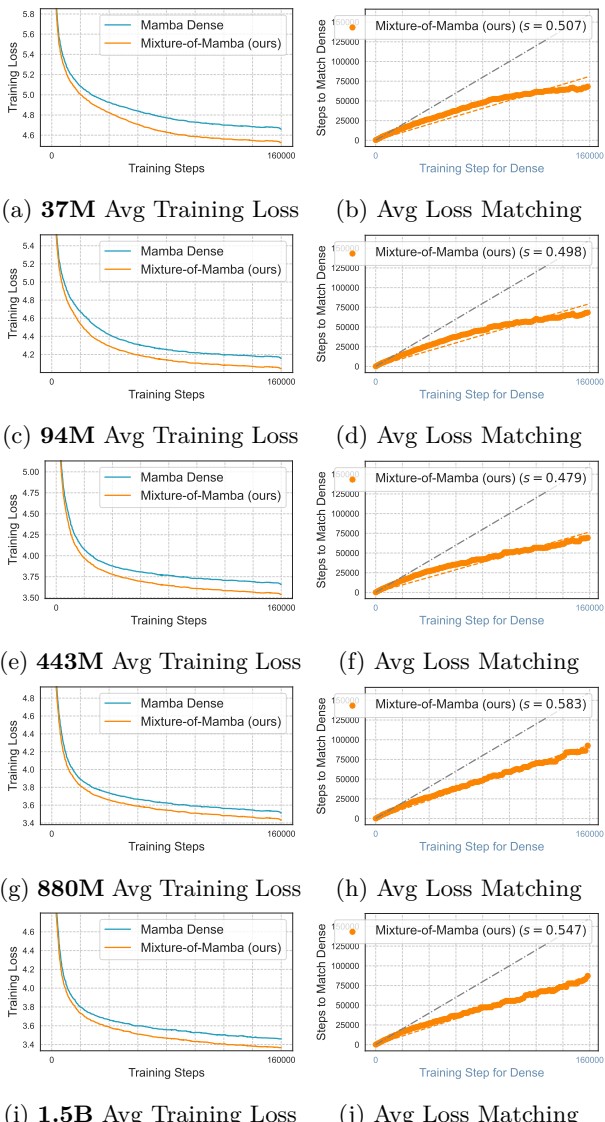

(a) **37M** Avg Training Loss    (b) Avg Loss Matching

(c) **94M** Avg Training Loss    (d) Avg Loss Matching

(e) **443M** Avg Training Loss    (f) Avg Loss Matching

(g) **880M** Avg Training Loss    (h) Avg Loss Matching

(i) **1.5B** Avg Training Loss    (j) Avg Loss Matching

Figure 10: **Average training loss and step matching plots across model scales in the Chameleon setting.** Results are shown for Mixture-of-Mamba and Mamba Dense across five model scales: **37M**, **94M**, **443M**, **880M**, and **1.5B**. **(a, c, e, g, i)** Average training loss (across image and text modalities) demonstrates consistent reductions for Mixture-of-Mamba (orange), achieving lower loss values compared to Mamba Dense (cyan) at all model scales. **(b, d, f, h, j)** Average loss matching plots highlight that Mixture-of-Mamba reaches the same loss values at earlier training steps compared to Mamba Dense, reflecting improved training efficiency. Overall, Mixture-of-Mamba consistently reduces average training loss and achieves more efficient convergence across all model scales in the Chameleon setting.

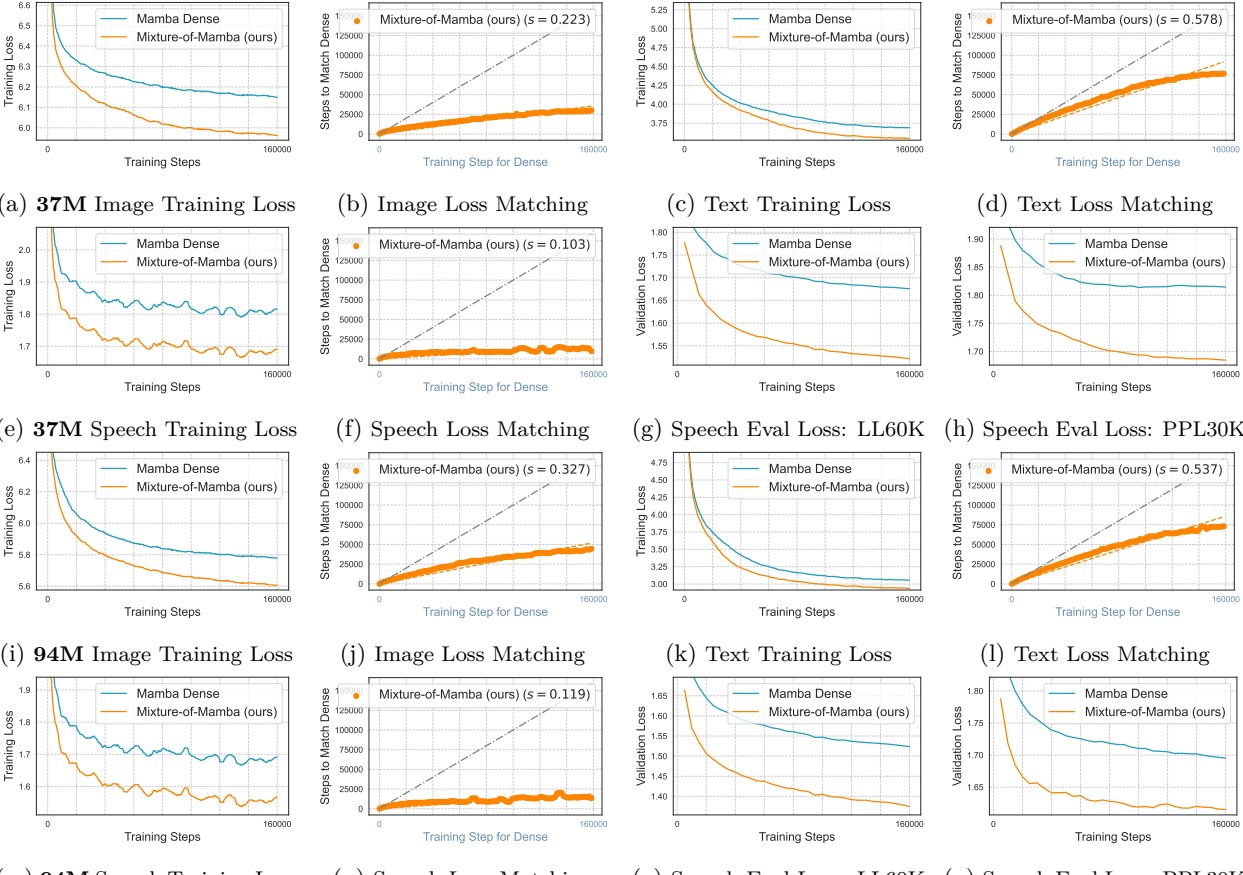

(a) **37M** Image Training Loss  (b) Image Loss Matching  (c) Text Training Loss  (d) Text Loss Matching

(e) **37M** Speech Training Loss  (f) Speech Loss Matching  (g) Speech Eval Loss: LL60K  (h) Speech Eval Loss: PPL30K

(i) **94M** Image Training Loss  (j) Image Loss Matching  (k) Text Training Loss  (l) Text Loss Matching

(m) **94M** Speech Training Loss  (n) Speech Loss Matching  (o) Speech Eval Loss: LL60K  (p) Speech Eval Loss: PPL30K

Figure 11: **Training and evaluation losses for image, text, and speech modalities (37M and 94M scales) in the Chameleon+Speech setting.** Results are reported for Mixture-of-Mamba and Mamba Dense. **(a, e, i)** Image training loss demonstrates that Mixture-of-Mamba (orange) achieves consistently lower loss compared to Mamba Dense (cyan). **(b, f, j)** Image loss matching highlights Mixture-of-Mamba's ability to reach the same loss values at earlier training steps, showing improved training efficiency. **(c, g, k)** Text training loss shows competitive results for Mixture-of-Mamba, improving over Mamba Dense. **(d, h, l)** Text loss matching confirms Mixture-of-Mamba's ability to reach the same loss values at earlier training steps, showing improved training efficiency. **(e, m)** Speech training loss highlights significant improvements in speech modality performance. **(f, n)** Speech loss matching shows efficient learning dynamics for Mixture-of-Mamba. **(g, o)** Speech evaluation loss on LL60K confirms notable performance gains, and **(h, p)** Speech evaluation loss on PPL30K further highlights the efficiency of Mixture-of-Mamba.

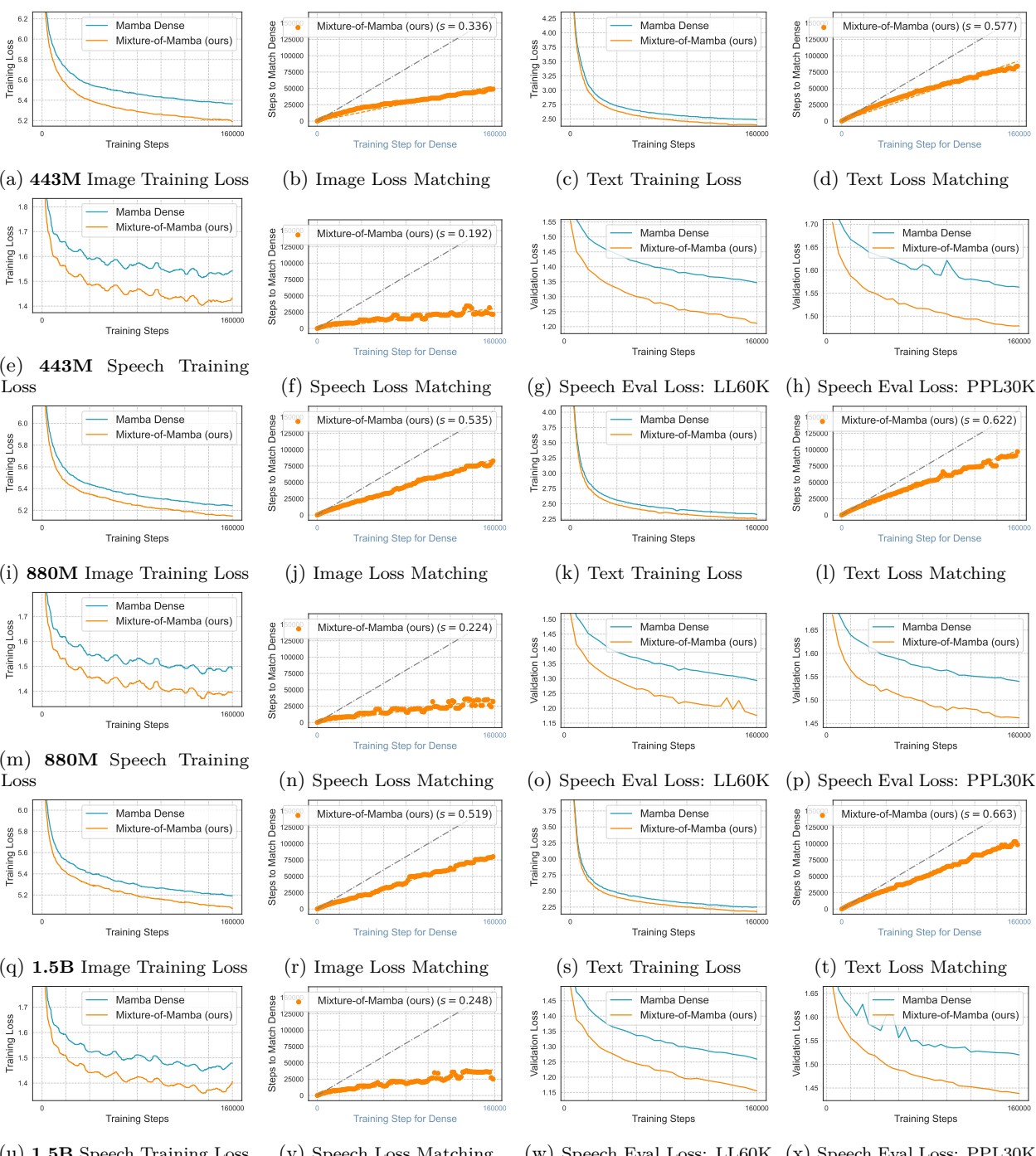

Figure 12: **Training and evaluation losses for image, text, and speech modalities (443M, 880M, and 1.5B scales) in the Chameleon+Speech setting.** Results are reported for Mixture-of-Mamba and Mamba Dense. **(a, i, q)** Image training loss demonstrates that Mixture-of-Mamba (orange) consistently outperforms Mamba Dense (cyan) across larger scales. **(b, j, r)** Image loss matching highlights improved training efficiency for Mixture-of-Mamba, reaching the same loss values at earlier training steps. **(c, k, s)** Text training loss shows Mixture-of-Mamba achieving better performance. **(d, l, t)** Text loss matching further demonstrates efficient learning dynamics. **(e, m, u)** Speech training loss confirms substantial gains for Mixture-of-Mamba in the speech modality, consistent across model scales. **(f, n, v)** Speech loss matching illustrates the improved efficiency of Mixture-of-Mamba across scales. **(g, o, w)** Speech evaluation loss on LL60K highlights consistent improvements, while **(h, p, x)** Speech evaluation loss on PPL30K demonstrates notable gains and efficient performance across scales.

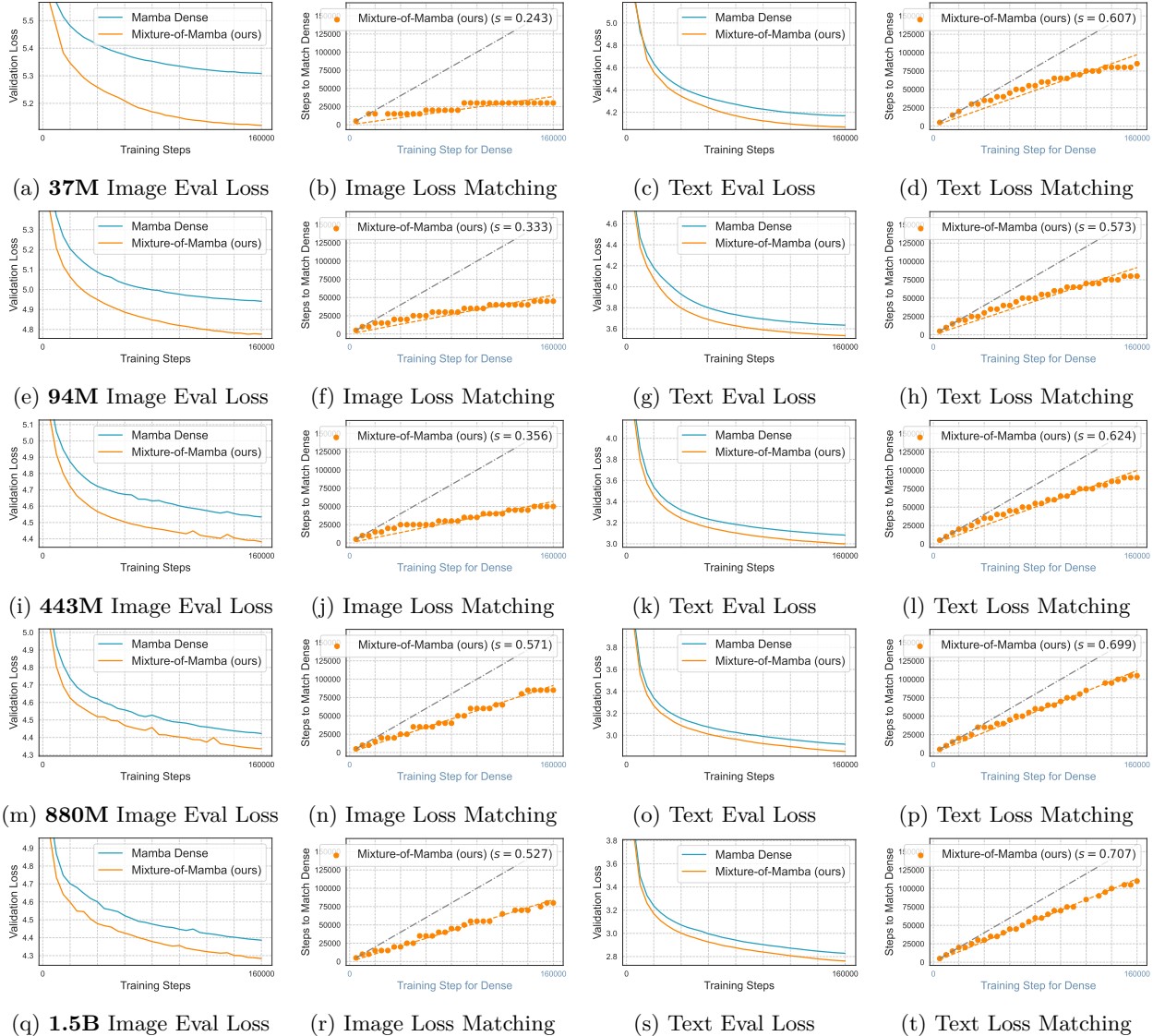

(a) **37M** Image Eval Loss    (b) Image Loss Matching    (c) Text Eval Loss    (d) Text Loss Matching

(e) **94M** Image Eval Loss    (f) Image Loss Matching    (g) Text Eval Loss    (h) Text Loss Matching

(i) **443M** Image Eval Loss    (j) Image Loss Matching    (k) Text Eval Loss    (l) Text Loss Matching

(m) **880M** Image Eval Loss    (n) Image Loss Matching    (o) Text Eval Loss    (p) Text Loss Matching

(q) **1.5B** Image Eval Loss    (r) Image Loss Matching    (s) Text Eval Loss    (t) Text Loss Matching

Figure 13: **Training and validation losses for image and text modalities across model scales in the Chameleon+Speech setting evaluated on the Obelisc dataset.** Results are shown for Mixture-of-Mamba and Mamba Dense across five model scales: **37M**, **94M**, **443M**, **880M**, and **1.5B**. **(a, e, i, m, q)** Image evaluation loss demonstrates consistent gains for Mixture-of-Mamba (orange) over Mamba Dense (cyan), even with the inclusion of the speech modality. **(b, f, j, n, r)** Image loss matching shows that Mixture-of-Mamba reaches the same loss values at earlier training steps compared to Mamba Dense, highlighting improved efficiency. **(c, g, k, o, s)** Text evaluation loss indicates consistent reductions for Mixture-of-Mamba relative to Mamba Dense across all scales. **(d, h, l, p, t)** Text loss matching illustrates that Mixture-of-Mamba reaches the same loss values at earlier training steps compared to Mamba Dense, maintaining its efficiency in the text modality. Overall, Mixture-of-Mamba achieves consistent improvements in both image and text modalities while maintaining its efficiency, even with the addition of the **speech modality**. These results confirm the robustness of Mixture-of-Mamba in multi-modal settings.

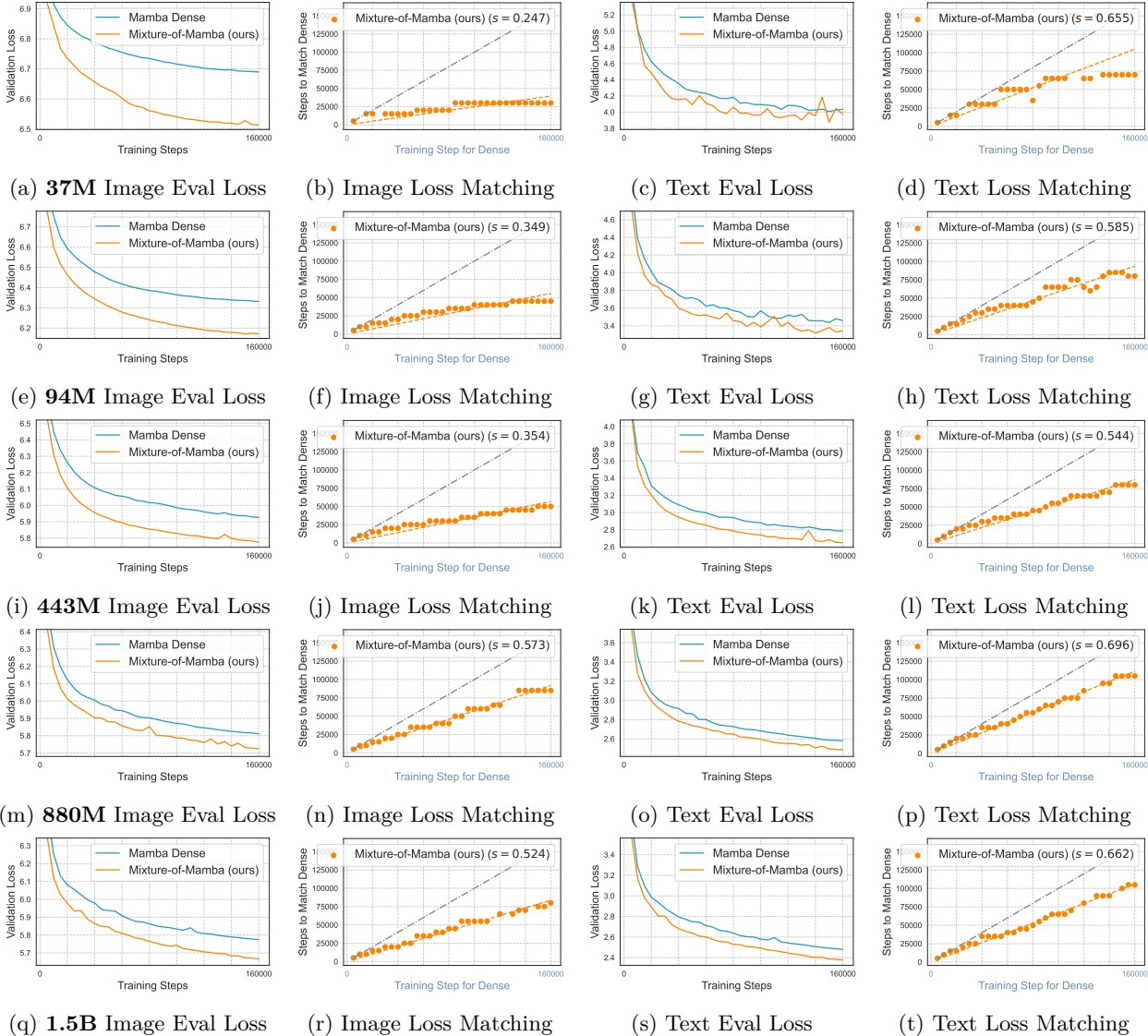

Figure 14: **Training and validation losses for image and text modalities across model scales in the Chameleon+Speech setting evaluated on the Shutterstock dataset.** Results are shown for Mixture-of-Mamba and Mamba Dense across five model scales: **37M**, **94M**, **443M**, **880M**, and **1.5B**. **(a, e, i, m, q)** Image evaluation loss demonstrates consistent gains for Mixture-of-Mamba (orange) over Mamba Dense (cyan), even with the inclusion of the speech modality. **(b, f, j, n, r)** Image loss matching shows that Mixture-of-Mamba reaches the same loss values at earlier training steps compared to Mamba Dense, highlighting improved efficiency. **(c, g, k, o, s)** Text evaluation loss indicates consistent reductions for Mixture-of-Mamba relative to Mamba Dense across all scales. **(d, h, l, p, t)** Text loss matching illustrates that Mixture-of-Mamba reaches the same loss values at earlier training steps compared to Mamba Dense, maintaining its efficiency in the text modality. Overall, Mixture-of-Mamba achieves consistent improvements in both image and text modalities while maintaining its efficiency, even with the addition of the **speech modality**. These results confirm the robustness of Mixture-of-Mamba in multi-modal settings.

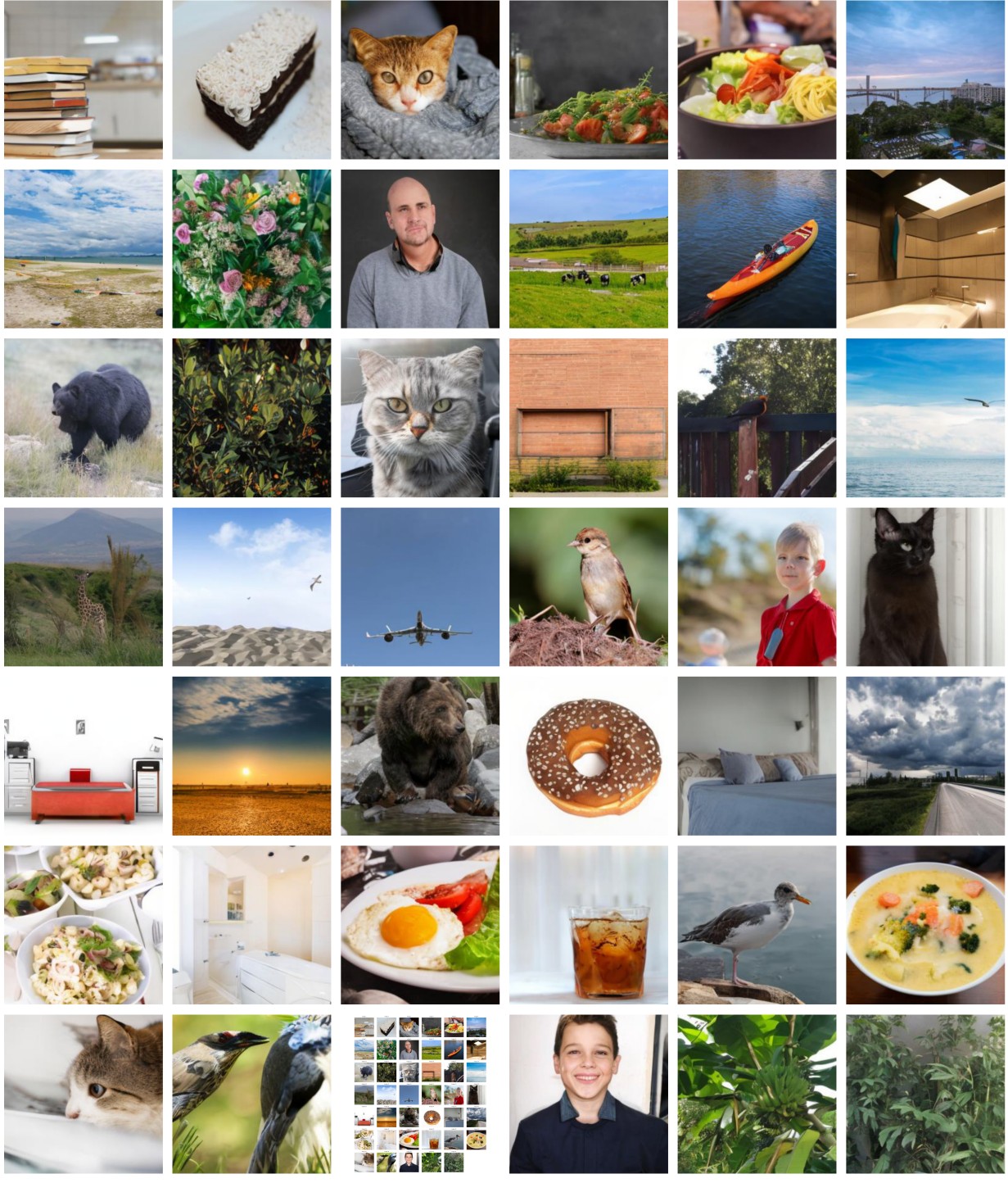

Figure 15: To demonstrate image quality, we include qualitative samples of Mixture-of-Mamba 1B generated images on COCO 2014.

