# OpenReview forum: "Mixture-of-Mamba: Enhancing Multi-Modal State-Space Models with Modality-Aware Sparsity"
_TMLR — Under review for TMLR_

### Review · Reviewer_Ngy4 · 2026-04-30

**Summary Of Contributions:**

The paper proposes Mixture-of-Mamba (MoM), a modality-aware sparse extension of the Mamba state-space model for multimodal pretraining. Its main contribution is to move sparsity inside the core Mamba block rather than applying MoE-style sparsity only to surrounding MLP layers. Specifically, the method decouples several Mamba projection components into modality-specific parameters for text, image, and speech tokens, allowing the model to specialize computation by modality while retaining the linear-time advantages of SSMs. The paper evaluates this idea across multiple multimodal settings, including Transfusion-style text-image training with diffusion loss, Chameleon-style discrete text-image modeling, and a three-modality text-image-speech setup. Across these settings, MoM generally matches or improves dense Mamba losses while using substantially fewer FLOPs, and the ablation study suggests that jointly decoupling input, intermediate, and output projections gives stronger gains than decoupling components individually.

Strength:
- Proposes a clear and novel architectural idea
- Provides broad empirical validation across multiple settings
- Includes useful ablations showing that jointly decoupling multiple Mamba projections gives stronger gains than decoupling individual components alone

Weaknesses:
- Evaluation relies heavily on training and validation loss, with limited downstream task evaluation beyond image generation
- Comparisons with larger multimodal models are not fully controlled due to differences in scale, data, and training setup
- The paper reports FLOPs but gives less evidence on wall-clock speed, memory usage, or deployment cost
- Reproducibility may be limited because some datasets, tokenizers, or training resources appear proprietary or difficult to access

**Audience:**

Yes

**Audience Explanation:**

Researchers working on efficient sequence modeling, state-space models, multimodal pretraining, and sparse architectures may found this work interesting. Its central finding, that modality-aware sparsity can be inserted directly into Mamba’s state-space projections and yield consistent efficiency and performance gains across text, image, and speech settings, is relevant to ongoing work on scalable alternatives to Transformer-based multimodal models.

**Broader Impact Concerns:**

The paper already includes an Impact Statement, and I do not see major broader-impact concerns beyond those typically associated with more efficient multimodal generative models.

**Claims And Evidence:**

Yes

**Claims Explanation:**

The authors provide experiments across Transfusion, Chameleon, and three-modality text-image-speech settings, report consistent loss and FLOP improvements over dense Mamba, include scalability results up to 1.5B parameters, and provide ablations showing the benefit of jointly decoupling projections.

**Requested Changes:**

- Provide stronger evidence that the reported FLOP reductions translate into practical efficiency gains.
- Expand downstream evaluation beyond loss curves and MS-COCO FID.
- Improve reproducibility details, especially for proprietary datasets, the in-house speech tokenizer, preprocessing, and training recipes.
- Include more discussion of limitations.

---

### Review · Reviewer_XsBc · 2026-06-08

**Summary Of Contributions:**

**Summary**

The Mixture-of-Mamba paper proposes an architectural extension to State-Space Models (SSMs) that integrates modality-specific "experts" into a unified framework. By leveraging the sequence-modeling capabilities of Mamba, the authors introduce a modality-aware parameter selection mechanism, where a modal mask is applied at the embedding layer to route inputs through specialized weight sets. Experimental results demonstrate that the model achieves comparable training loss while reducing computational cost by 35–65% in FLOPS compared to dense Mamba. Furthermore, the architecture maintains full compatibility with existing optimized Mamba CUDA kernels, ensuring efficient deployment.

**Strengths**

1. Significant Training Efficiency: The architecture demonstrates a substantial improvement in training efficiency, achieving comparable loss with 35–65% fewer FLOPS than dense Mamba.

2. Extensive Empirical Validation: The paper provides a comprehensive evaluation, featuring a wide variety of experiments, detailed ablations, and a large number of reported metrics, which collectively demonstrate the model's behavior across different scales and training configurations.


**Weaknesses**

1. Routing Mechanism & Inference Flexibility: Sections 2.2 and 2.3 rely on fixed input structures—Transfusion uses delimiters, and the image encoder employed in Chameleon experiments produces a fixed sequence length of 1024 tokens. As there is no explicit router or classifier layer, it appears the model requires prior knowledge of the data structure to generate the 'modal mask.' If the data structure must remain known and fixed throughout, what is the fundamental architectural advantage of this 'merged' Mamba approach over simply training two separate specialized models? Could author clarify that if MoM offers any mechanism to handle dynamically interleaved, unlabeled streams, or is it strictly dependent on external metadata provided by the data loader?

2. Performance Metrics & Benchmarking: The reported 'Performance gain in percentage' across all tables could be misleading, as these figures represent relative percentage improvements in **training/validation loss** only, which are nonlinearly correlated to real downstream performance metrics like accuracy or F1 scores. Furthermore, the downstream comparison in Table 5 and the discussion in Section 3.5 are inconsistent: As Sec 3.5 highlights a gain against the 7B Chameleon while omitting a direct comparison against the more relevant Transfusion baseline at matched scales. Could author provide a direct 1B MoM vs. 1B Transfusion comparison (or similarly a 7B comparison) to clarify whether the efficiency gains of MoM does not involve a fundamental trade-off in generative quality?

**Audience:**

Yes

**Audience Explanation:**

Mamba is a popular architecture that has been studied and utilized heavily. Any improvement in training of deployment would benefit the community.

**Broader Impact Concerns:**

No concerns.

**Claims And Evidence:**

Yes

**Claims Explanation:**

The improvement in training efficiency is clear and convincing. But author would need to address the "efficiency-accuracy trade-off concern" mentioned in Weaknesses 2.

**Requested Changes:**

Please address the questions raised in Weaknesses above.

---

> ### Author Response · Authors · 2026-07-21
>
> We thank Reviewer XsBc for the careful review and for recognizing the training-efficiency gains and empirical breadth. We address Reviewer XsBc's concerns below.
>
>
> ---
>
> ## Weakness 1: Routing mechanism and inference flexibility
>
> > **Reviewer comment:** “Sections 2.2 and 2.3 rely on fixed input structures—Transfusion uses delimiters, and the image encoder employed in Chameleon experiments produces a fixed sequence length of 1024 tokens. As there is no explicit router or classifier layer, it appears the model requires prior knowledge of the data structure to generate the modal mask. If the data structure must remain known and fixed throughout, what is the fundamental architectural advantage of this merged Mamba approach over simply training two separate specialized models? Could the authors clarify whether MoM offers any mechanism to handle dynamically interleaved, unlabeled streams, or is it strictly dependent on external metadata provided by the data loader?”
>
> Thank you for the comment. We have revised the manuscript to state how the modality mask is constructed. MoM does not use a learned router or a classifier inside the model. The data loader and sequence-construction pipeline provide a token-level modality label, and that label selects the corresponding modality-specific projection path.
>
> In the Chameleon and speech settings, text, image, and speech are represented as discrete tokens with known token sources. The loader can determine the modality from the token ID and the tokenizer stream used to produce it: text tokens are routed to the text projections, image-code tokens to the image projections, and speech-code tokens to the speech projections. In the Transfusion setting, `<Begin of Image>` and `<End of Image>` delimit image spans inside an interleaved text-image sequence, so tokens inside the span are routed as image tokens and tokens outside the span are routed as text tokens.
>
> This routing rule does not require one fixed sequence layout. The Chameleon image encoder used in our experiments produces 1,024 image tokens, but that length is a property of the tokenizer configuration, not an architectural constraint of MoM. The same mask construction applies to variable-length segments, multiple image spans, and dynamically interleaved sequences, provided that the sequence carries token-level modality labels. What MoM does not attempt to solve is blind segmentation of a completely unlabeled raw stream; that setting would require an upstream modality detector and is outside the scope of the paper.
>
> The advantage over training separate specialized models is that MoM remains one sequence model. The recurrent state transition and Conv1D path are shared across the full interleaved stream, so text, image, and speech tokens can condition on one another inside a single Mamba backbone. Only the projections that directly map token features are modality-specific. Separate models would duplicate the backbone and require an additional fusion mechanism to exchange cross-modal information; MoM instead keeps cross-modal sequence modeling shared while decoupling the modality-dependent projection maps.
>
> We have revised the paper to make this scope explicit: MoM uses deterministic routing from available token-level modality labels; it supports arbitrarily interleaved labeled multimodal token streams; and it does not claim to infer modality labels from unlabeled raw inputs.

---

### Review · Reviewer_iSPW · 2026-07-10

**Summary Of Contributions:**

This paper proposes Mixture-of-Mamba (MoM), which introduces modality-aware sparsity directly inside the Mamba block by replacing several projection layers with modality-specific parameters. Unlike previous sparse Mamba variants that mainly apply sparsity outside the SSM block (e.g., in MLP layers), the proposed method performs modality-specific specialization within the state-space projections themselves. The paper evaluates the approach on three multimodal pretraining settings (Transfusion, Chameleon, and a three-modality text-image-speech setup) and reports improved training efficiency together with comparable or better validation performance. The paper also includes ablation studies and a downstream image generation evaluation using FID.

**Audience:**

Yes

**Audience Explanation:**

This topic concerns state-space-models, which should be a relevant topic for the TMLR community.

**Broader Impact Concerns:**

I do not see broader impact concerns.

**Claims And Evidence:**

No

**Claims Explanation:**

The empirical ev\aluation is fairly comprehensive. The authors evaluate multiple model sizes, multiple multimodal training settings, and provide ablation studies on different projection layers. The reported improvements are generally consistent across experiments, suggesting that the proposed architecture is effective under the evaluated settings.

However, I do not think the evidence is fully convincing for some of the paper's stronger claims.

First, the paper repeatedly emphasizes that introducing modality-aware sparsity inside the Mamba block is the key contribution, but there is very little analysis explaining why this design works. The experiments mainly show performance improvements, but do not provide evidence that the modality-specific projections actually learn different functionality or reduce modality interference. I am wondering whether some representation analysis or parameter similarity analysis could help support this claim.

Second, the comparison with related sparse Mamba architectures is insufficient. The paper discusses MoE-Mamba and BlackMamba extensively and argues that these methods are orthogonal, but they remain among the most relevant baselines. Without direct experimental comparisons under similar parameter and compute budgets, it is difficult to judge the practical advantage of the proposed approach. I believe the authors could at least provide evidence showing the proposed method can work with MoE-Mamba or BlackMamba to make them better, in order to support the claim of orthogonality.

Finally, the downstream evaluation is still somewhat limited. Since the motivation is efficient multimodal foundation models, it would be more convincing to also evaluate multimodal understanding tasks such as VQA, captioning, or retrieval, rather than relying primarily on pretraining losses and a single image generation benchmark.

Overall, I think the experimental evidence supports that the method is effective, but it is less convincing regarding the novelty of the design and the reasons behind the observed improvements.

**Requested Changes:**

1. Please include comparisons with more closely related sparse Mamba architectures (e.g., MoE-Mamba or BlackMamba) under comparable scale, or show that the propose method can be combined with these sparse atchitectures. These seem to be the most natural baselines.
2. I would like to see more analysis explaining why modality-specific projections improve performance. For example, do different modality-specific projections actually learn noticeably different representations? Some visualization or representation analysis would strengthen the paper considerably.
3.The downstream evaluation could be strengthened. Besides image generation FID, I think evaluating multimodal understanding tasks (e.g., VQA, image captioning, retrieval, or instruction-following benchmarks) would better demonstrate that the gains transfer beyond pretraining loss.
4. I am wondering why the state transition matrix and Conv1D layers remain shared while the projection layers are decoupled. The paper gives an intuitive explanation, but some empirical validation of this design choice would make the argument stronger
5. Since many reported improvements are relatively modest, please consider reporting results over multiple random seeds or including standard deviations to demonstrate that the improvements are statistically reliable.